# Controlled SDEs for Long-Horizon Motion Generation under Latent Decision Uncertainty

**Han Zhang** [1 2 3]  **Nenggan Zheng** [1 3 4 5 6]

## Abstract

Long-horizon motion prediction under external commands is challenged by latent decision uncertainty, where the internal states governing future behavior are unobservable and evolve stochastically over time. This issue is particularly pronounced in biological agents, whose motion trajectories reflect decision-making processes rooted in underlying cognitive states. To address these challenges, we propose CogSDE, a formulation of a controlled stochastic differential equation (SDE) for modeling instruction-driven latent decision dynamics. The drift term in the SDE incorporates a dual-channel control modulation mechanism, enabling external commands to modulate the evolution of latent states. The diffusion term employs a state-dependent operator to model intrinsic uncertainty in latent decision dynamics. Furthermore, we establish dissipativity-based mean-square boundedness for the latent decision dynamics. Experiments demonstrate that CogSDE consistently improves predictive accuracy in long-horizon motion generation. Importantly, predicted trajectories remain well aligned with control commands over extended horizons, a property widely recognized as challenging in long-horizon motion prediction.

## 1. Introduction

Predicting the long-horizon motion of biological agents under external control commands is critical for intervention-aware decision support (Finn et al., 2016). By predicting how different commands will influence future motion, control models can prevent unsafe behavioral deviations and collisions, thereby enabling more robust decision-making under deep uncertainty (Mayne et al., 2000). This task arises in a wide range of applications, including interactive human behavior modeling (Park et al., 2023), human motion generation (Zhang et al., 2024), and behavioral prediction based on brain–computer interfaces (Metzger et al., 2023).

A fundamental challenge in command-driven motion prediction lies in understanding the internal decision-making processes of biological agents. These processes evolve over time and mediate how external commands are translated into actions, resulting in latent decision uncertainty. Because such internal states are only partially observable, their stochastic evolution can accumulate over time and significantly influence long-horizon motion outcomes (Friston et al., 2017; Kaelbling et al., 1998). Although previous discriminative (Shi et al., 2023; Wang et al., 2024) and generative motion (Wen et al., 2023; Xu et al., 2024) models have addressed some aspects of this problem, they typically lack an explicit representation of how time-varying commands causally intervene in latent decision dynamics over long horizons. Therefore, effective modeling requires capturing continuous-time decision dynamics, intrinsic uncertainty, and time-varying control interventions within a unified framework.

To address these requirements, stochastic differential equation (SDE) provides a natural framework for modeling latent decision dynamics. However, most existing formulations are limited in modeling command-driven decision dynamics. First, control signals are often incorporated as static conditioning variables rather than as explicit, time-varying interventions that actively modulate the evolution of latent states (Kidger et al., 2020; 2021). Consequently, these models struggle to represent how discrete-time commands induce structured transitions in internal decision states over time. Second, uncertainty in many latent dynamical approaches is not treated as an intrinsic component of decision dynamics.

---

[1]Qiushi Academy for Advanced Studies (QAAS), Zhejiang University, Zhejiang 310058, China [2]College of Computer Science and Technology, Zhejiang University, Zhejiang 310058, China [3]Shanghai Institute for Advanced Study of Zhejiang University (SIAS), Shanghai 201203, China [4]State Key Laboratory of Brain-Machine Intelligence, Zhejiang University, Hangzhou, Zhejiang 310007, China [5] Collaborative Innovation Center for Artificial Intelligence by MOE and Zhejiang Provincial Government (ZJU), Hangzhou 310007, China [6]School of Computer and Information Engineering, Bengbu University, Bengbu 233030, China. Correspondence to: Nenggan Zheng <zng@cs.zju.edu.cn>.

*Proceedings of the 43rd International Conference on Machine Learning*, Seoul, South Korea. PMLR 306, 2026. Copyright 2026 by the author(s).

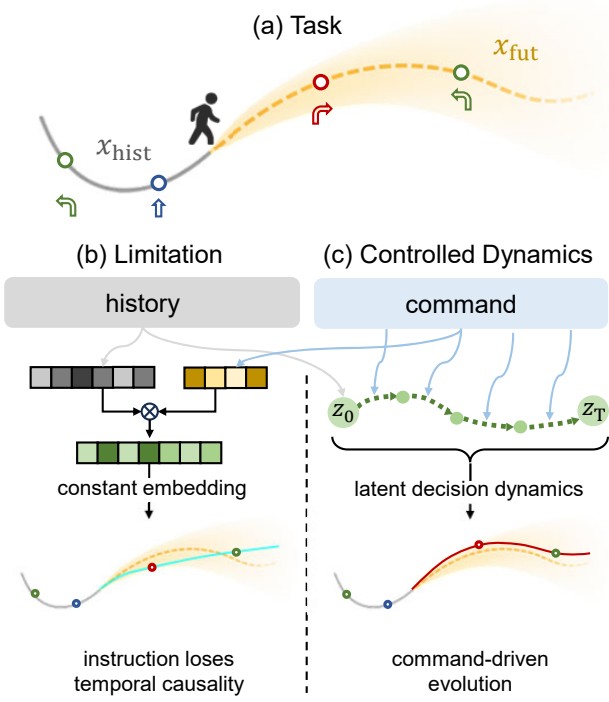

*Figure 1.* Overview of instruction-driven long-horizon motion prediction. (a) Given past trajectories and external commands, the goal is to generate future motion, although the underlying biological decision process remains unobserved. (b) Existing methods typically treat commands as static conditioning variables throughout the prediction process. (c) In this work, the latent decision process is modeled by CogSDE, whose evolution is continuously modulated over time by external commands, thereby enabling bounded and controllable motion generation.

In these methods, stochasticity is typically introduced for regularization or smoothing purposes rather than to capture uncertainty arising from partially observed and evolving cognitive processes (Rubanova et al., 2019). Together, these limitations hinder the modeling of how time-varying commands causally influence both the trajectory and uncertainty of latent decision states, motivating the development of a unified framework that explicitly integrates continuous-time dynamics, intrinsic stochasticity, and intervention-aware control.

As illustrated in Figure 1, we propose CogSDE, a novel formulation of a controlled SDE designed to model command-driven, long-horizon latent decision dynamics. CogSDE formulates the evolution of internal decision states as a continuous-time stochastic process, providing a principled representation for long-horizon decision dynamics under partial observability. Within this framework, external commands are incorporated through a dual-channel control modulation mechanism acting on the drift term, allowing control signals to intervene in the latent dynamics as time-varying modulators. Additionally, the diffusion term employs a state-dependent stochastic operator to model intrinsic uncertainty

in latent decision dynamics, enabling uncertainty to persist and propagate over extended horizons. Building on this controlled stochastic formulation, CogSDE couples latent decision dynamics with motion generation in a well-behaved manner. We further derive an analytical mean-square boundedness result that characterizes the long-horizon behavior of the controlled latent dynamics under standard dissipativity assumptions.

Our contributions are threefold:

- We propose CogSDE, a controlled SDE framework for long-horizon, command-driven motion prediction. CogSDE enables uncertainty-aware and controlled motion generation by integrating intrinsic stochasticity with control-modulated latent dynamics.

- We formulate a command-driven, long-horizon motion prediction problem that unifies forecasting and conditional generation under uncertainty. This formulation captures time-specific control interventions and latent decision uncertainty. It is evaluated on a rat locomotion dataset collected under controlled, command-driven settings, along with established public benchmarks.

- We provide a mean-square boundedness analysis that characterizes the long-horizon motion of the proposed framework. Under a standard one-sided dissipativity assumption, we derive a sufficient condition for the boundedness of the latent stochastic dynamics in the presence of persistent control inputs.

## 2. Related Work

**Deterministic and Autoregressive Motion Prediction** Discriminative models formulate behavior prediction as supervised regression, mapping historical observations to a single future trajectory using deterministic sequence models (Shi et al., 2022; Liu et al., 2024; Shi et al., 2023; Wang et al., 2024; Shi et al., 2024). Autoregressive models extend this paradigm by sequentially generating future poses or tokens, often borrowing language-modeling mechanisms to support long-horizon prediction (Li et al., 2020; Jiang et al., 2023; Seff et al., 2023; Hong et al., 2025). In both models, external commands are incorporated as static context or token-level bias, leaving the underlying state-transition mechanism unchanged and preventing commands from reshaping the evolution of latent behavioral states (Benidis et al., 2022). Consequently, long-horizon prediction relies on repeated pointwise extrapolation, resulting in brittle trajectories whose errors compound and whose performance rapidly degrades under distributional shifts or delayed command effects.

**Generative Models** Generative models formulate motion prediction by sampling future trajectories from a learned

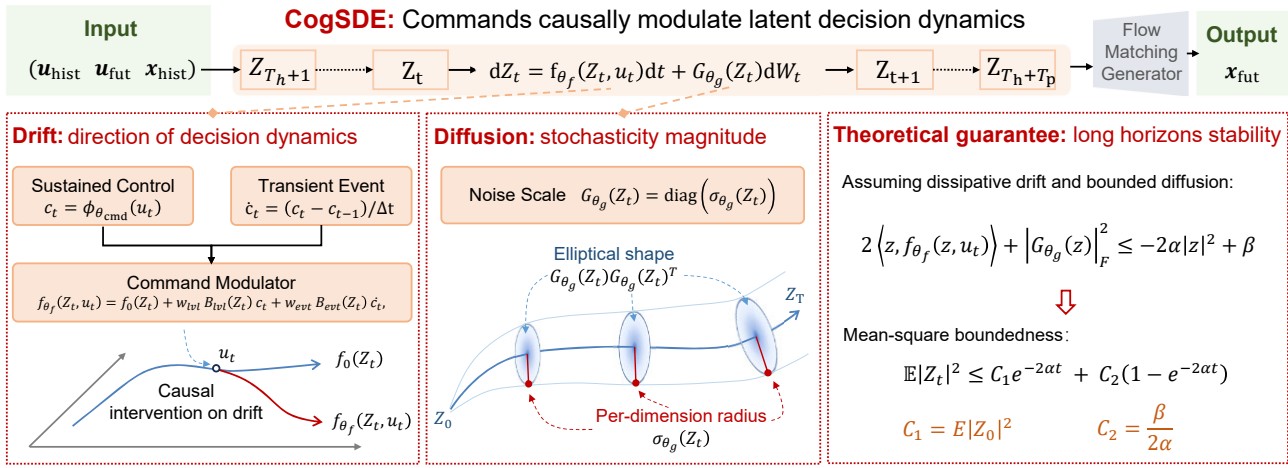

Figure 2. Overview of CogSDE for command-driven long-horizon motion prediction. Commands causally intervene on the latent drift to modulate decision dynamics from the initial state $Z_{T_h}$. The drift is decomposed into sustained and transient control channels, capturing persistent and abrupt command effects, respectively. Intrinsic uncertainty is modeled by a state-dependent diagonal diffusion that regulates the magnitude of stochastic perturbations. Under assumptions of dissipative drift and bounded diffusion, we analyze the mean-square boundedness of the latent dynamics. Finally, latent states are decoded by a flow-matching generator to produce future motion.

conditional distribution, enabling explicit modeling of multi-modal outcomes (Harshvardhan et al., 2020; Pinheiro Cinelli et al., 2021). In diffusion-based models, conditioning signals guide the denoising process by biasing the score or noise prediction, thereby influencing both the direction and progression of generation (Song et al.; Pan et al., 2024; Cao et al., 2024; Yang et al., 2025). Flow-matching and rectified-flow methods incorporate conditions by modulating the velocity field that transports samples along probability paths, directly shaping the direction of latent evolution (Lipman et al., 2023; Gat et al., 2024; Hu et al., 2024). While many diffusion-based and flow-based models incorporate external inputs through guidance or conditioning mechanisms (Dhariwal & Nichol, 2021; Ajay et al.; Karras et al., 2022), their influence on long-horizon behavior is typically indirect, as the conditioning signal primarily biases the generation process rather than explicitly parameterizing latent state evolution. In contrast, our approach focuses on a dual-channel drift design that directly modulates latent dynamics, making it particularly well-suited for modeling sparse and abrupt command effects in long-horizon motion prediction.

**Latent Dynamical Systems** Latent dynamical models represent motion through the continuous-time evolution of hidden states, providing a compact abstraction of internal behavioral dynamics beyond observable trajectories (Kim et al., 2018). Controlled latent models extend this framework by incorporating external inputs or control signals into the state evolution, enabling time-varying modulation of latent dynamics (Watter et al., 2015; Cuchiero et al., 2020; Kidger et al., 2020). Furthermore, stochastic latent models introduce randomness via latent SDE, capturing intrinsic uncertainty and multimodal evolution within a continuous state space (Park et al., 2021; Peters et al., 2022; Hasan

et al., 2021). However, in sequence prediction settings, existing latent dynamical approaches typically treat external instructions as conditioning variables rather than as causal interventions on latent dynamics, which limits their ability to model instruction-driven state transitions and counterfactual long-horizon behaviors (Li et al., 2023; Chapfuwa et al., 2022).

## 3. Methods

### 3.1. Problem Setup

We study the problem of command-driven motion prediction under partial observation, where only external motion states and control commands are observable, while the internal states that govern future motion evolution remain unobserved. Given a historical motion sequence along with both historical and future sequences of external commands, the goal is to predict future motion over a fixed horizon. Formally, the input consists of a historical pose sequence $x_{1:T_h} \in \mathbb{R}^{T_h \times V \times C}$, where each frame contains $V$ keypoints represented in a $C$-dimensional coordinate space. The model is also provided with a per-timestep encoded command sequence $u_{1:T_h+T_p} \in \mathbb{R}^{(T_h+T_p) \times d_u}$, which is temporally aligned with both the observed history and the prediction horizon. Let $x_{\text{hist}} = x_{1:T_h}$, $x_{\text{fut}} = x_{T_h+1:T_h+T_p}$, and $u = u_{1:T_h+T_p}$. With these definitions, the objective can be formulated as modeling the conditional distribution:

$$p_\Theta(x_{\text{fut}} \mid x_{\text{hist}}, u). \qquad (1)$$

To account for the unobserved decision-making process, we reformulate Eq. (1) as a latent-variable dynamical model. Specifically, a latent trajectory $Z_{\text{fut}} = Z_{T_h+1:T_h+T_p}$ represents the unobserved decision states that govern motion

evolution under external commands. Let $h = (x_{\text{hist}}, u)$ denote the observed history-command context:

$$p_\Theta(x_{\text{fut}} \mid h) = \int p_{\theta_{\text{dyn}}}(Z_{\text{fut}} \mid h) p_{\theta_{\text{dec}}}(x_{\text{fut}} \mid Z_{\text{fut}}) \, dZ_{\text{fut}}. \tag{2}$$

The latent process is initialized from observed history through a deterministic encoder. Let $u_{\text{hist}} = u_{1:T_h}$:

$$Z_{T_h} = \phi_{\theta_{\text{enc}}}(x_{\text{hist}}, u_{\text{hist}}). \tag{3}$$

### 3.2. Controlled SDE for Latent Decision Dynamics

To model command-driven motion evolution under partial observation, we introduce a latent stochastic process that captures unobserved internal states governing future dynamics, as illustrated in Figure 2. The latent state is designed to aggregate historical motion information while evolving under the influence of external commands.

Let $Z_t \in \mathbb{R}^{d_z}$ denote the latent state at continuous time $t$. Its evolution is governed by a controlled SDE of the form:

$$dZ_t = f_{\theta_f}(Z_t, u_t) \, dt + G_{\theta_g}(Z_t) \, dW_t, \tag{4}$$

where $W_t$ is a standard Brownian motion, $f_{\theta_f}$ denotes the drift function modulated by the control input $u_t$, and $G_{\theta_g}$ denotes the diffusion function characterizing the intrinsic stochasticity of biological agents.

The controlled drift term $f_{\theta_f}(Z_t, u_t)$ defines how external commands causally influence the latent dynamics, rather than serving as static conditioning variables. This formulation allows commands to modulate the instantaneous rate of latent state evolution at each time step. In contrast, the diffusion term $G_{\theta_g}(Z_t)$ captures unobserved variability that cannot be explained solely by deterministic dynamics or external inputs.

Following Eq. (3), we initialize $Z_{T_h}$ from the observed history $(x_{\text{hist}}, u_{\text{hist}})$ and evolve it with Eq. (4). Future motion is modeled through a sample-based generative process conditioned on latent dynamics. Specifically, the latent state modulates the vector field of the flow-matching model, thereby shaping the evolution from noise to future motion. This framework separates latent decision dynamics from observable motion generation, allowing command-dependent evolution and stochastic uncertainty to be explicitly modeled in continuous time.

### 3.3. Command-Modulated Drift

We now specify the structure of the command-driven drift term $f_{\theta_f}(Z_t, u_t)$, which governs the evolution of latent dynamics under external commands. To model both the persistence of past commands and the impact of newly introduced commands, we introduce a dual-channel control modulation mechanism that explicitly balances sustained control effects with transient command variations. Under this formulation, the drift is defined as

$$f_{\theta_f}(Z_t, u_t) = f_0(Z_t) + w_{\text{lvl}} \mathcal{B}_{\text{lvl}}(Z_t) c_t + w_{\text{evt}} \mathcal{B}_{\text{evt}}(Z_t) \dot{c}_t, \tag{5}$$

where $f_0(Z_t)$ denotes the control-independent latent dynamics evaluated pointwise from the current state $Z_t$. The continuous control representation $c_t = \phi_{\theta_{\text{cmd}}}(u_t)$ is obtained by encoding discrete commands using a learnable command encoder, and $\dot{c}_t \approx (c_t - c_{t-1})/\Delta t$ denotes the temporal difference of $c_t$. The weights $w_{\text{lvl}}$ and $w_{\text{evt}}$ balance the relative influence of sustained command persistence and transient command updates, thereby mediating the trade-off between stimulus retention and forgetting.

The level channel $\mathcal{B}_{\text{lvl}}(Z_t) c_t$ models the instantaneous influence of the current command value $c_t$ on latent dynamics. This term captures sustained control effects.

The event channel $\mathcal{B}_{\text{evt}}(Z_t) \dot{c}_t$ captures the effect of rapid command changes by operating on the temporal difference $\dot{c}_t$. This term enables the model to respond to abruptly changing commands.

The state-dependent operators $\mathcal{B}_{\text{lvl}}(\cdot)$, and $\mathcal{B}_{\text{evt}}(\cdot)$ are parameterized using a shared regime-based mixture structure:

$$\mathcal{B}_{\text{lvl}}(Z_t) = \sum_{i=1}^{S} \pi_i(Z_t) B_i^{\text{lvl}}, \quad \mathcal{B}_{\text{evt}}(Z_t) = \sum_{i=1}^{S} \pi_i(Z_t) B_i^{\text{evt}}, \tag{6}$$

where $\boldsymbol{\pi}(Z_t) \in \Delta^{S-1}$ denotes regime weights satisfying $\pi_i(Z_t) \geq 0$ and $\sum_i \pi_i(Z_t) = 1$. This regime-based mixture is adopted as a shared parameterization for both drift and diffusion components.

### 3.4. Diffusion for Intrinsic Uncertainty

The diffusion term is modeled as a diagonal, state-dependent operator:

$$G_{\theta_g}(Z_t) = \text{diag}\big(\sigma_{\theta_g}(Z_t)\big), \tag{7}$$

where $\sigma_{\theta_g}(Z_t) \in \mathbb{R}_+^{d_z}$ controls the scale of stochastic perturbations along each latent dimension. Unlike constant diffusion assumptions, this formulation allows the level of uncertainty to adapt to the current latent state, enabling the model to represent time-varying stochastic effects.

The effective diffusion scale is calculated as a regime-weighted aggregate:

$$\sigma_{\theta_g}(Z_t) = \sum_{i=1}^{S} \pi_i(Z_t) \sigma_i(Z_t), \tag{8}$$

where $\sigma_i(Z_t)$ is a positive, state-dependent diffusion scale formed by modulating a regime-specific base scale with a

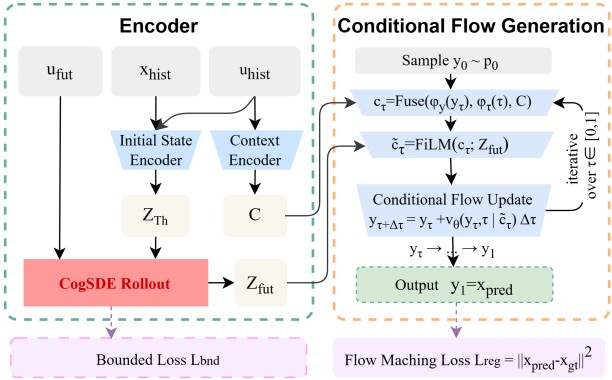

*Figure 3.* Computational pipeline of CogSDE. Historical motion and commands are encoded into context tokens $C$ and the initial latent state $Z_{T_h}$. The CogSDE rollout produces $Z_{\text{fut}}$ based on future commands, which modulates the conditional flow decoder via FiLM. The decoder iteratively maps $y_0$ to $y_1 = x_{\text{pred}}$, with a trajectory prediction loss applied to the output and an optional boundedness regularizer applied to the latent rollout.

shared, latent-state-dependent residual. Additionally, the same regime weights $\pi_i(Z_t)$ used in the drift term are also applied in the diffusion term. This shared parameterization ensures that stochastic variability remains consistent with the underlying latent regimes.

### 3.5. Computational Pipeline and Discretization

Figure 3 illustrates the end-to-end computation, which consists of history encoding, CogSDE rollout, and conditional flow generation. First, the history encoder extracts context tokens $C$ from $x_{\text{hist}}$, and the initial latent state $Z_{T_h}$ is obtained by Eq. (3). Second, although the controlled SDE is formulated in continuous time, its implementation relies on Euler–Maruyama discretization with a finite step size:

$$Z_{k+1} = Z_k + f_{\theta_f}(Z_k, u_k)\Delta t + G_{\theta_g}(Z_k)\sqrt{\Delta t}\epsilon_k, \quad (9)$$

where $\epsilon_k \sim \mathcal{N}(0, I)$, and $Z_k$ denotes the approximation of $Z(t_k)$ at $t_k = k\Delta t$. Applying Eq. (9) over $k = T_h, \ldots, T_h + T_p - 1$ yields $Z_{\text{fut}}$.

Finally, the latent rollout is used to condition the flow-matching decoder. For a noisy flow state $y_\tau$ at flow time $\tau$, we construct

$$c_\tau = \text{Fuse}(\phi_y(y_\tau), \phi_\tau(\tau), C). \quad (10)$$

The full latent rollout $Z_{\text{fut}}$ modulates this base condition through FiLM (Perez et al., 2018):

$$\tilde{c}_\tau = \text{FiLM}(c_\tau; Z_{\text{fut}}). \quad (11)$$

The modulated condition $\tilde{c}_\tau$ parameterizes the conditional velocity field, and future motion is generated by integrating

$$y_{\tau+\Delta\tau} = y_\tau + v_\theta(y_\tau, \tau \mid \tilde{c}_\tau)\Delta\tau \quad (12)$$

from noise to data space, with the terminal state $y_1$ taken as $x_{\text{pred}}$. The model is trained end-to-end with the trajectory prediction objective:

$$\mathcal{L}_{\text{reg}} = \|x_{\text{pred}} - x_{\text{gt}}\|^2. \quad (13)$$

### 3.6. Mean-Square Boundedness for Long-Horizon Prediction

To analyze the long-horizon motion of the controlled neural SDE, we examine the mean-square evolution of the latent SDE in Eq. (4) by studying the second moment $Y(t) := \mathbb{E}|Z_t|^2$.

A tractable analysis of this quantity requires a structural assumption on the controlled drift and diffusion terms. Intuitively, this assumption requires the latent dynamics be globally contractive in the mean-square sense, up to a constant offset induced by control inputs and stochastic noise. Formally, we introduce a one-sided dissipativity assumption (Müller et al., 2014):

**Assumption 3.1** (One-sided Dissipativity)**.** There exist constants $\alpha > 0$ and $\beta > 0$ such that, for all latent state $z \in \mathbb{R}^{d_z}$ and bounded control inputs satisfying $\|u_t\| \leq U_{\max}$, the drift and diffusion functions satisfy

$$2\langle z, f_{\theta_f}(z, u_t)\rangle + \|G_{\theta_g}(z)\|_F^2 \leq -2\alpha\|z\|^2 + \beta, \quad (14)$$

where $\alpha$ reflects the one-sided dissipativity of the controlled drift, and $\beta$ upper bounds the aggregate energy contribution from bounded control inputs and stochastic noise. We use lowercase $z$ for pointwise structural assumptions and uppercase $Z_t$ for the latent stochastic process.

Under the above assumption, we adopt the quadratic Lyapunov function $V(Z_t) = |Z_t|^2$ to analyze the mean-square boundedness of the latent process (Hosoe & Hagiwara, 2021). Applying Itô's lemma to $V(Z_t)$ yields a differential inequality for the evolution of the second moment $\mathbb{E}|Z_t|^2$, which leads to the following mean-square boundedness result (Itô).

**Theorem 3.1** (Mean-Square Boundedness of the Latent SDE)**.** *Consider the SDE* (4) *with drift and diffusion given by* (5) *and* (7)*, and assume Assumption 3.1 holds and the control sequence satisfies* $|u_t| \leq U_{\max}$*. Then the second moment satisfies, for all* $t \geq 0$,

$$\mathbb{E}|Z_t|^2 \leq C_1 e^{-2\alpha t} + C_2\big(1 - e^{-2\alpha t}\big), \quad (15)$$

*where*

$$C_1 = \mathbb{E}|Z_0|^2, \quad C_2 = \frac{\beta}{2\alpha}. \quad (16)$$

*Here, $C_1$ represents the initial latent energy, while $C_2$ gives the asymptotic mean-square bound. This bound is determined by the balance between the dissipative rate $\alpha$ and the residual energy contribution $\beta$ from bounded commands and diffusion.*

*Table 1.* Performance comparison under the command-driven motion prediction setting on the Rat Locomotion and BABEL datasets, evaluated using minimum ADE.

| METHOD | RAT LOCOMOTION ($\text{ADE}_{\min}$) | | | | | | BABEL ($\text{ADE}_{\min}$) | | | | | |
|---|---|---|---|---|---|---|---|---|---|---|---|---|
| | 10 | 20 | 30 | 40 | 50 | 60 | 10 | 20 | 30 | 40 | 50 | 60 |
| TIMEXER | 1.69 | 3.11 | 4.75 | 6.43 | 8.04 | 9.57 | 0.010 | 0.017 | 0.026 | 0.035 | 0.044 | 0.054 |
| SLD-HMP | 0.47 | 1.02 | 1.70 | 2.42 | 3.15 | 3.89 | 0.006 | 0.009 | 0.013 | 0.016 | 0.020 | 0.023 |
| DIFFSTG | 5.24 | 6.12 | 7.46 | 9.04 | 10.68 | 12.29 | 0.244 | 0.249 | 0.256 | 0.264 | 0.272 | 0.280 |
| MOFLOW | 0.86 | 1.73 | 2.77 | 3.92 | 5.13 | 6.34 | 0.005 | 0.009 | 0.013 | 0.017 | 0.021 | 0.025 |
| **COGSDE** | **0.44** | **0.90** | **1.50** | **2.21** | **2.98** | **3.79** | **0.003** | **0.006** | **0.009** | **0.012** | **0.015** | **0.018** |

**Proof sketch.** Applying Itô's lemma to $V(Z) = |Z|^2$ gives $\frac{d}{dt}\mathbb{E}\|Z_t\|^2 = \mathbb{E}\big[2\langle Z_t, f_{\theta_f}(Z_t, u_t)\rangle + \|G_{\theta_g}(Z_t)\|_F^2\big]$. Under Assumption 3.1, the right hand side is upper bounded by $-2\alpha\,\mathbb{E}\|Z_t\|^2 + \beta$, leading to a linear differential inequality. Applying Grönwall's inequality gives the bound in Eq. (15). A complete derivation, including the detailed Itô expansion and constant definitions, is provided in Appendix A.

This result shows that the controlled latent SDE remains mean-square bounded under the stated dissipativity assumption. Specifically, the contribution of the initial latent energy decays exponentially at a rate of $2\alpha$, and the residual energy introduced by bounded commands and diffusion determines the asymptotic upper bound $C_2 = \beta/(2\alpha)$. Therefore, $\alpha$ controls how fast the latent process forgets its initial condition, and $C_2$ characterizes the limiting energy scale of the latent rollout. We note that the analysis is derived for the continuous-time SDE, whereas training and inference use Euler–Maruyama discretization with a finite step size. A fully discrete-time analysis would require additional smoothness and step-size assumptions. Therefore, we treat this result as a continuous-time stability characterization and empirically examine the stability of discretized rollouts in Section 4.4.

To connect the analysis with learning, we incorporate the dissipativity condition as a soft penalty on the latent rollout. At each step $k$, we evaluate

$$r_k = 2\langle Z_k, f_{\theta_f}(Z_k, u_k)\rangle + \|G_{\theta_g}(Z_k)\|_F^2 + 2\alpha\|Z_k\|^2, \quad (17)$$

and penalize violations of $r_k \le \beta$ by

$$\mathcal{L}_{\text{bnd}} = \frac{\sum_k w_k\,\eta\,\text{softplus}((r_k - \beta)/\eta)}{\sum_k w_k}. \quad (18)$$

Here, $w_k \ge 0$ emphasizes later rollout steps, and $\tau$ controls the smoothness of the penalty. This term is added to the training objective using a warmup-and-ramp schedule for its weight $\lambda_{\text{bnd}}$, allowing it to act as a gentle stability bias rather than a hard constraint. Empirically, moderate weights reduce dissipativity violations and improve rollout stability, whereas excessively large weights can over-constrain the latent dynamics and degrade prediction accuracy.

## 4. Experiments

We conduct a series of experiments, including comparative evaluations against representative baselines to examine the predictive performance of our framework, and systematic ablation studies to analyze the effects of its key components. In addition, we perform targeted empirical analyses to examine whether the learned dynamics align with the boundedness analysis. Further experimental results and analyses are provided in Appendix C. Implementation and training details are available in Appendix D.

### 4.1. Experiment Setup

**Dataset** We evaluate CogSDE primarily on a real-world rat locomotion dataset collected under controlled electrical stimulation, where future motion is causally driven by explicit commands. Each sample consists of historical 2D keypoint trajectories, temporally aligned control commands (history and future), and the corresponding ground-truth future motion. Detailed data collection, preprocessing, and ethical approvals are provided in Appendix B. We further report results on the BABEL human motion dataset (Punnakkal et al., 2021) to assess cross-domain generalization, treating action labels as external commands following prior work.

**Evaluation** Prediction accuracy is evaluated using minimum ADE and average ADE, which measure best-case and distribution-level pose accuracy, respectively.

### 4.2. Long-Horizon Motion Prediction Performance

**Baselines** We compare CogSDE with representative baselines encompassing both deterministic forecasting and generative motion modeling paradigms to ensure a fair and comprehensive evaluation.

Specifically, we include transformer-based predictors TimeXer (Wang et al., 2024) as deterministic baselines, which treat commands as exogenous inputs without explicitly modifying state-transition dynamics, reflecting a common design choice in long-horizon forecasting.

For generative modeling, we compare our approach against

*Figure 4.* Visualization of five representative trajectory prediction cases from the Rat Locomotion dataset. Gray curves represent the observed history, and orange dashed curves indicate the ground-truth future trajectories. Colored curves correspond to predictions from different methods, including TimeXer (blue), SLD-HMP (purple), MoFlow (green), and our method, CogSDE (red). Stimulation commands and their categories (circle) are marked along the trajectories. Each case highlights the model's ability to respond to time-varying stimulation and generate long-horizon motion accordingly.

*Table 2.* Distribution-level pose accuracy evaluated using average ADE on the Rat Locomotion and BABEL datasets.

| METHOD | RAT LOCOMOTION (ADE$_{avg}$) | | | | | | BABEL (ADE$_{avg}$) | | | | | |
|---|---|---|---|---|---|---|---|---|---|---|---|---|
| | 10 | 20 | 30 | 40 | 50 | 60 | 10 | 20 | 30 | 40 | 50 | 60 |
| SLD-HMP | 2.01 | 3.90 | 5.83 | 7.72 | 9.50 | 11.18 | 2.175 | 3.830 | 4.861 | 5.478 | 5.858 | 6.121 |
| MOFLOW | 1.84 | 4.26 | 7.30 | 10.68 | 14.13 | 17.41 | 0.010 | 0.020 | 0.030 | 0.041 | 0.052 | 0.062 |
| **COGSDE** | **0.81** | **1.96** | **3.65** | **5.74** | **8.05** | **10.42** | **0.005** | **0.011** | **0.019** | **0.027** | **0.035** | **0.042** |

diffusion-based DiffSTG (Wen et al., 2023) and flow-based MoFlow (Fu et al., 2025) methods. These models capture stochastic motion evolution but do not inherently support explicit command-driven control. Following standard practice, we evaluate these models in their original, unconditioned form, as naive command injection was empirically found to degrade performance and compromise model validity.

To bridge this gap and facilitate a direct comparison with command-aware generative modeling, we also include SLD-HMP (Xu et al., 2024), which integrates high-level commands via structured latent directions. All methods are trained and evaluated using identical data splits and horizons (30-frame history, 60-frame prediction). For multimodal generative models, we report both best-of-K (K=20) and distribution-level metrics.

**Overall Prediction Performance** Table 1 summarizes minimum ADE on two datasets with distinct motion dynamics. On the Rat Locomotion dataset, CogSDE achieves the lowest error across all horizons, with the most significant improvements observed in 40–60 frames. It consistently outperforms the command-aware baseline, SLD-HMP. These results support our claim that modeling command interventions as drift modulation, combined with stochastic diffusion to capture latent decision uncertainty.

On BABEL, CogSDE remains the top-performing method across all horizons, demonstrating that control-aware stochastic dynamics generalize beyond animal locomotion. Notably, performance gaps widen as the horizon length increases, with methods incorporating stochastic latent dynamics becoming more robust. CogSDE consistently achieves the lowest error over long horizons.

**Multi-Modal Performance** Table 2 presents the average ADE for multimodal generative models on Rat Locomotion and BABEL datasets. As a distribution-level metric, average ADE reflects the overall quality of the sampled trajectories and is sensitive to poorly regulated multimodal generation.

Across both datasets, CogSDE consistently achieves the lowest average ADE at all horizons, indicating more controlled sampling behavior during long-horizon prediction. In contrast, SLD-HMP and MoFlow exhibit a faster increase in average ADE as the horizon lengthens, suggesting a decline in distribution-level quality. These results show that control-modulated latent SDE dynamics in CogSDE enable better-regulated multimodal prediction over extended horizons.

**Visualization** We visualize representative prediction results with a focus specific to each dataset. For Rat Locomotion, we present multiple trajectory cases to illustrate the high global uncertainty caused by sparse stimulation commands. For BABEL, where trajectories are more regular, we instead visualize a representative pose sequence to emphasize fine-grained limb dynamics.

As shown in Figure 4, CogSDE consistently captures command-aligned turning trends over long horizons, producing smoother and more stable trajectories than deterministic (TimeXer) and flow-based (MoFlow) baselines. Highly scattered predictions from diffusion-based models are omitted for visual clarity. Additionally, as illustrated in Figure 5, CogSDE generates coordinated arm and leg motions that closely follow the ground-truth pose evolution, whereas SLD-HMP exhibits limited arm articulation and fails to achieve the intended transition.

*Table 3.* Ablation study on the Rat Locomotion dataset. We evaluate the effects of drift modulation and diffusion modeling using minimum ADE and average ADE.

| ID | DRIFT | DIFFUSION | $ADE_{min}$ | | | | | | $ADE_{avg}$ | | | | | |
|---|---|---|---|---|---|---|---|---|---|---|---|---|---|---|
| | | | 10 | 20 | 30 | 40 | 50 | 60 | 10 | 20 | 30 | 40 | 50 | 60 |
| M0 | | | 0.86 | 1.73 | 2.77 | 3.92 | 5.13 | 6.34 | 1.84 | 4.26 | 7.30 | 10.68 | 14.13 | 17.41 |
| M1 | ✓ | | **0.42** | 0.82 | 1.42 | 2.20 | 3.13 | 4.12 | 0.95 | 2.34 | 4.49 | 7.26 | 10.35 | 13.47 |
| M2 | | ✓ | 0.44 | **0.82** | **1.35** | **2.01** | **2.76** | **3.55** | 0.97 | 2.33 | 4.41 | 7.00 | 9.81 | 12.60 |
| M3 | ✓ | ✓ | 0.44 | 0.90 | 1.50 | 2.21 | 2.98 | 3.79 | **0.81** | **1.96** | **3.65** | **5.74** | **8.05** | **10.41** |

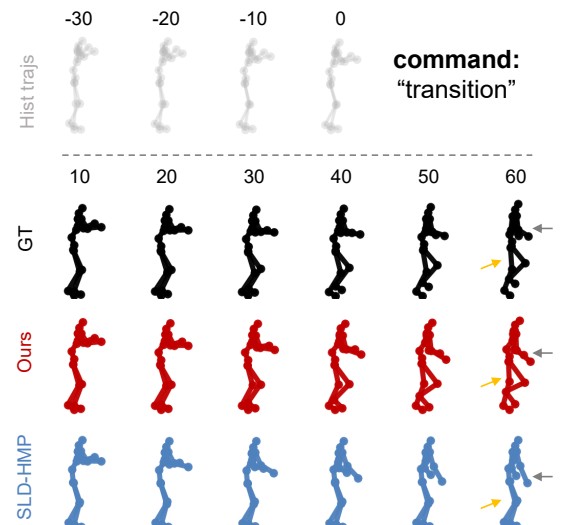

*Figure 5.* Qualitative visualization of pose sequence prediction on the BABEL dataset under a transition command. The top row shows the historical poses, followed by the ground-truth future poses (GT). Predictions from CogSDE (red) and SLD-HMP (blue) are shown at selected future time steps. The visualization emphasizes fine-grained limb dynamics and long-horizon pose evolution under instruction-driven settings.

## 4.3. Ablation Studies

We conduct ablation studies to analyze the contributions of key components in the proposed framework. Specifically, we compare four model variants that progressively incorporate design choices related to latent conditioning and temporal fusion. All variants are trained and evaluated under identical settings.

**M0: Base MoFlow.** The base MoFlow model is used as the generative backbone, modeling future motion through flow-based dynamics without any command-aware modulation or stochastic latent augmentation.

**M1: MoFlow + Drift Modulation.** This variant augments MoFlow with a command-modulated drift term, enabling deterministic, time-dependent control over the generative dynamics while keeping the noise structure unchanged.

**M2: MoFlow + Stochastic Diffusion.** In this setting, stochastic latent noise is introduced into the dynamics to

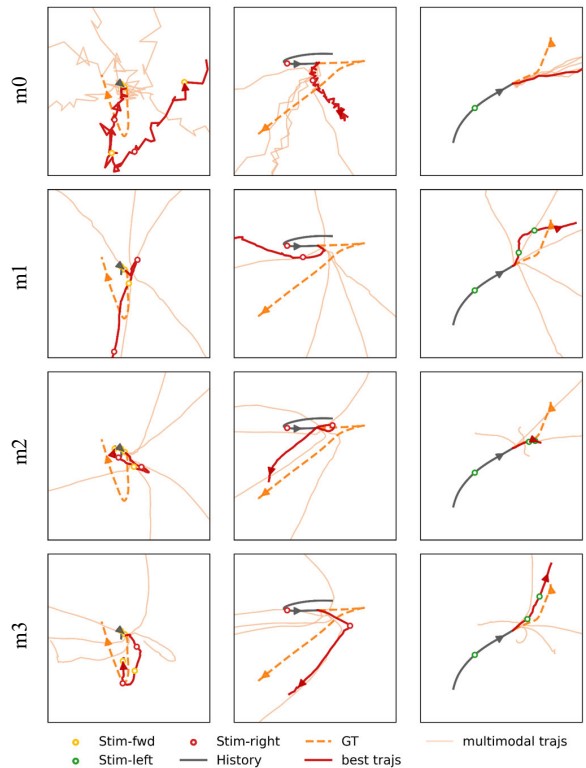

*Figure 6.* Trajectory predictions across ablation variants M0–M3. The rows, from top to bottom, correspond to models with progressively added components. Gray curves represent the observed history, orange dashed curves indicate ground-truth future trajectories, red curves show the best predicted trajectory, and thin, light-colored solid lines depict multiple sampled trajectories.

model intrinsic uncertainty and multimodality, without explicit command modulation of the drift.

**M3: Full Model (CogSDE).** The full model combines command-modulated drift and stochastic latent noise, forming a controlled stochastic dynamical system for command-conditioned motion generation.

**Quantitative Analysis** Table 3 shows that adding either drift modulation (M1) or stochastic diffusion (M2) reduces the minimum ADE compared to the baseline (M0). Moreover, enabling both components simultaneously (M3) results in the lowest average ADE across all horizons. This finding indicates that deterministic control modulation and stochastic

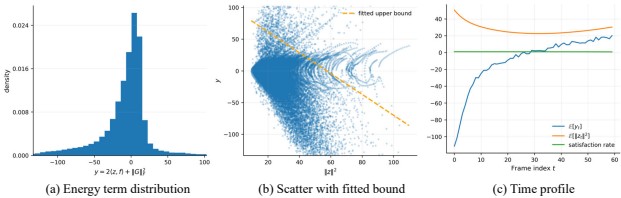

(a) Energy term distribution   (b) Scatter with fitted bound   (c) Time profile

*Figure 7.* Post-hoc empirical analysis of a dissipativity proxy on the test set. Panels (a)–(c) summarize the distribution of the dissipativity energy term, an empirical upper bound fitted in distribution, and time-resolved statistics over long-horizon rollouts, respectively.

*Table 4.* Effect of boundedness regularization at horizon 60.

| $\lambda_{\mathrm{bnd}}$ | HORIZON | $\mathrm{ADE_{min}}$ | $\mathrm{FDE_{min}}$ | $\mathrm{ADE_{avg}}$ | $\mathrm{FDE_{avg}}$ |
|---|---|---|---|---|---|
| 0 | 60 | 3.79 | 6.54 | 10.42 | 24.27 |
| 0.001 | 60 | 3.65 | 6.31 | 12.37 | 27.90 |
| 0.003 | 60 | 3.67 | 6.37 | 11.48 | 24.99 |

uncertainty modeling are complementary and jointly essential for well-regulated, long-horizon multimodal prediction.

**Visualization** As illustrated in Figure 6, progressively adding drift and diffusion improves both the best trajectory alignment and the consistency of predictions, leading to more coherent long-horizon motion under commands.

### 4.4. Empirical Analysis of Long-Horizon Stability

To complement the analytical boundedness result in Section 3.6, we first perform a post-hoc empirical analysis to examine whether the learned latent dynamics exhibit behavior consistent with the one-sided dissipativity assumption on test trajectories.

$$y_t = 2\langle z_t, f_{\theta_f}(z_t, u_t)\rangle + |G_{\theta_g}(z_t)|_F^2. \tag{19}$$

As shown in Figure 7(a), the distribution of $y_t$ is predominantly negative, indicating a contractive tendency in the learned drift, while allowing occasional violations due to stochasticity. Figure 7(b) plots $y_t$ against the latent energy $|z_t|^2$. An empirical upper bound of the form $y_t \leq -2\alpha|z_t|^2 + \beta$ is fitted by performing robust linear regression on the upper envelope of the scatter distribution to cover a fixed proportion of samples ($p = 0.95$). This suggests that the learned dynamics are broadly consistent with a dissipativity-type inequality in distribution. Time-resolved statistics in Figure 7(c) show that the mean latent energy remains bounded over long horizons without divergence, and the fraction of states satisfying the proxy inequality remains consistently high throughout the rollout.

We evaluate the boundedness loss $\mathcal{L}_{\mathrm{bnd}}$ as a stability-oriented regularizer linking the boundedness analysis to training. As shown in Table 4, sweeping $\lambda_{\mathrm{bnd}} \in \{0, 0.001, 0.003\}$ shows that a moderate weight $\lambda_{\mathrm{bnd}} = 0.001$ reduces $\mathrm{ADE_{min}}$ from 3.79 to 3.65, indicating im-

proved best-sample rollout quality. However, the average-sample metrics do not improve consistently, indicating that the regularizer may also affect the spread of generated samples. Overall, the sweep supports using the boundedness term as a mild stability bias rather than a strong constraint.

## 5. Conclusion

In this work, we propose CogSDE for command-driven long-horizon motion prediction in biological agents. By embedding external control commands and intrinsic randomness into continuous-time latent dynamics, CogSDE provides an interpretable mechanism for controlled, conditional motion generation. Furthermore, we derive a dissipativity-based mean-square boundedness result that offers theoretical insight into the long-horizon behavior of the proposed model. Extensive experimental evaluations demonstrate that CogSDE enables accurate and command-consistent long-horizon motion prediction under external control, highlighting controlled stochastic dynamics as an effective modeling framework for robust conditional generation.

Despite these results, several aspects remain open for further extension. First, the current rat experiments involve a limited number of subjects, making cross-subject generalization an important area for future research. Second, characterizing the admissible step-size range and smoothness conditions required for fully discrete-time boundedness remains a critical direction. Finally, extending CogSDE to handle higher-dimensional observations, such as video-level representations, would provide a more rigorous test of the framework's scalability.

## Acknowledgments

We gratefully acknowledge support from the Shanghai Key Technology Research and Development Program under Grant 25XF3200500, the Excellent Research Innovation Team of Anhui Provincial Education Department under Grant 2024AH010018, the Leading Goose R&D Program of Zhejiang Province under Grant 2025C02045 and 2026C02A1226, the National Natural Science Foundation of China under Grant T2293723, and the Natural Science Foundation of Zhejiang Province under Grant LZ24F020003. The authors also thank Prof. Junwen Mao for valuable support in manuscript preparation, and Manman Ji for assistance with experimental data collection.

## Impact Statement

This paper presents work whose goal is to advance the field of Machine Learning. There are many potential societal consequences of our work, none which we feel must be specifically highlighted here.

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

# A. Proof details of Theorem 3.1 (Mean-square boundedness).

Consider the latent stochastic process $\{Z_t\}_{t\geq 0}$ governed by

$$dZ_t = f_{\theta_f}(Z_t, u_t)\, dt + G_{\theta_g}(Z_t)\, dW_t, \tag{20}$$

and define the Lyapunov function $V(z) = \|z\|^2$.

Since $\nabla V(z) = 2z$ and $\nabla^2 V(z) = 2I$, applying Itô's formula yields

$$d\|Z_t\|^2 = 2\langle Z_t, f_{\theta_f}(Z_t, u_t)\rangle\, dt + \|G_{\theta_g}(Z_t)\|_F^2\, dt + 2\langle Z_t, G_{\theta_g}(Z_t)\, dW_t\rangle. \tag{21}$$

Under standard Lipschitz and linear-growth conditions on $f_{\theta_f}$ and $G_{\theta_g}$, the stochastic integral $\int_0^t \langle Z_s, G_{\theta_g}(Z_s)\, dW_s\rangle$ is a true martingale. Taking expectation on both sides of (21) gives

$$\frac{d}{dt}\mathbb{E}\|Z_t\|^2 = \mathbb{E}\left[2\langle Z_t, f_{\theta_f}(Z_t, u_t)\rangle + \|G_{\theta_g}(Z_t)\|_F^2\right]. \tag{22}$$

By Assumption 3.1, for all $z \in \mathbb{R}^{d_z}$ and bounded controls $\|u_t\| \leq U_{\max}$, there exist constants $\alpha > 0$ and $\beta \geq 0$ such that

$$2\langle z, f_{\theta_f}(z, u_t)\rangle + \|G_{\theta_g}(z)\|_F^2 \leq -2\alpha\|z\|^2 + \beta. \tag{23}$$

Applying this inequality pointwise to (22) yields the differential inequality

$$\frac{d}{dt}Y(t) \leq -2\alpha Y(t) + \beta, \qquad Y(t) := \mathbb{E}\|Z_t\|^2. \tag{24}$$

Solving the corresponding equality and applying the comparison principle, we obtain

$$\mathbb{E}\|Z_t\|^2 \leq \mathbb{E}\|Z_0\|^2 e^{-2\alpha t} + \frac{\beta}{2\alpha}\left(1 - e^{-2\alpha t}\right), \tag{25}$$

which proves (15). $\qquad\square$

*Remark* A.1 (On the validity of Assumption 3.1). Assumption 3.1 corresponds to a standard one-sided dissipativity condition commonly adopted in the boundedness analysis of stochastic dynamical systems. In our setting, this condition is facilitated by the model design: the drift is parameterized via a mixture-of-experts structure with bounded command embeddings, which constrains the energy injection from external inputs, while the diffusion term is diagonal and its magnitude is explicitly regulated. As a result, the aggregate contribution of control inputs and stochastic noise can be uniformly bounded by a constant $\beta$. We emphasize that Assumption 1 serves as a sufficient condition to mean-square boundedness, rather than a restrictive requirement.

# B. Dataset details

**Ethics and Animal Welfare Statement** This work involved animal subjects in its research. Approval of all ethical and experimental procedures and protocols was granted by the Laboratory Animal Welfare and Ethics Committee of Zhejiang University under LAWEC AP Code: ZJU20220260. All experiments were performed in accordance with the guidelines for the care and use of laboratory animals at Zhejiang University (ZJU, Hangzhou, China).

**Overview** The raw rat locomotion dataset consists of 36 trials collected from three rats under command-driven navigation settings. As illustrated in Figure 8(a), each trial records the rat's locomotion within a rectangular arena of approximately $2\,\mathrm{m} \times 2\,\mathrm{m}$, where the animal navigates from a designated start location toward a sequence of predefined checkpoints. Rat motion is recorded using a top-down camera at 18 frames per second. From the recorded videos, 2D skeletal keypoints are extracted using an in-house pose estimation pipeline and subsequently verified through manual inspection. Each trial has a variable duration, ranging from approximately $T_{\min}$ to $T_{\max}$ seconds.

As shown in Figure 8(b), control commands are delivered via an invasive brain–computer interface (BCI) integrated with a sensing–control microsystem, enabling direct electrical stimulation to elicit forward, left-turn, and right-turn locomotion

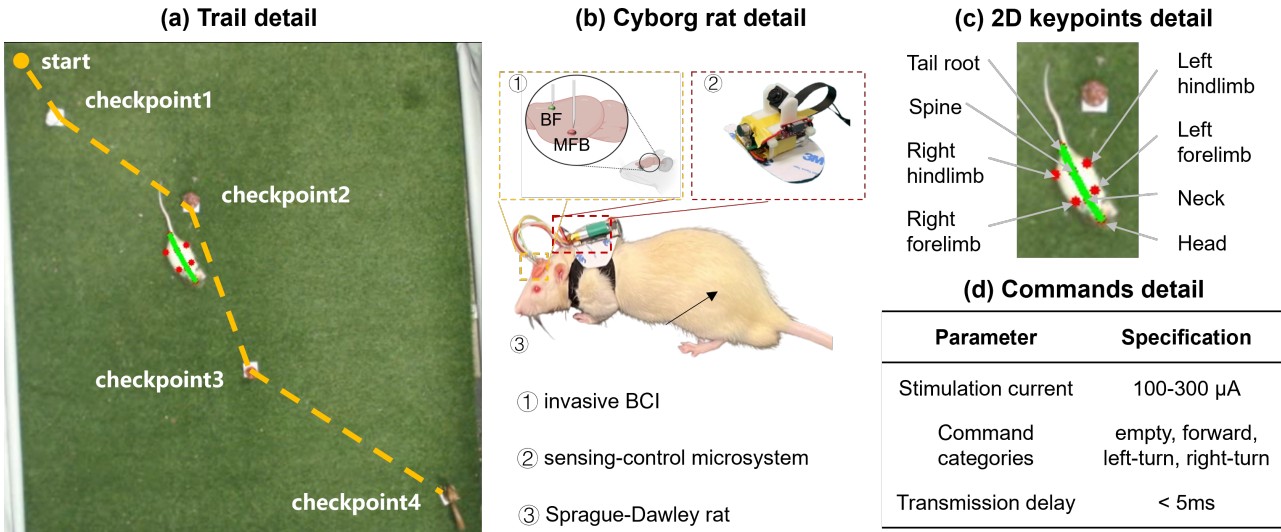

*Figure 8.* Rat locomotion dataset and command-driven navigation setup. (a) Example trial trajectory in a $2\,\mathrm{m} \times 2\,\mathrm{m}$ arena. (b) Cyborg rat platform with invasive brain–computer interface. (c) 2D skeletal keypoint representation. (d) Command specifications.

behaviors. Specifically, forward commands are delivered by stimulating the medial forebrain bundle (MFB), while left-turn and right-turn commands are elicited via stimulation of the left and right whisker-related cortical regions, respectively. Each trial contains synchronized 2D skeletal keypoints (Figure 8(c)) and discrete control instructions, including empty instructions indicating the absence of external control (Figure 8(d)). Together, these modalities form the raw input for instruction-conditioned rat locomotion modeling. This dataset provides a controlled yet biologically grounded testbed for studying instruction-conditioned locomotion prediction.

**Data preprocess** Based on the collected trials, we construct fixed-length motion sequences using a sliding-window strategy. Each sample consists of 30 frames of historical motion and 60 frames of future motion, resulting in sequences of 90 frames in total. The sliding window advances with a stride of 15 frames along each trial.

Each frame is represented by 8 skeletal keypoints, with each keypoint encoded by 2D coordinates. After preprocessing, this procedure yields a total of 6,471 motion equences with shape $(90, 8, 2)$. The resulting dataset is randomly split into training, validation, and test sets with a ratio of 70%, 20%, and 10%, respectively. The dataset is not pre-normalized. Instead, each model applies its own normalization strategy during training.

**Command Encoding and Temporal Alignment** Control commands are discrete categorical signals with four possible states: forward, left-turn, right-turn, and empty (no instruction). Each command is temporally aligned with the corresponding motion frames. Commands are delivered via electrical stimulation with fixed current amplitudes selected according to individual rats and experimental states, typically ranging from $100\,\mu$A to $300\,\mu$A, with each stimulation lasting approximately $100\,$ms. The transmission delay of electrical stimulation is negligible ($\leq 5\,$ms) relative to the motion sampling rate, while the subsequent biological response latency varies across individuals and states. Such response latency is treated as part of the intrinsic, unobservable decision-making dynamics and is therefore implicitly captured by the latent stochastic process.

During training, commands are encoded as bounded embeddings and provided as external control inputs to the model. Empty instructions indicate the absence of external control. For animal safety, no more than three stimulation commands are typically issued per second, resulting in sparse and short-duration control signals. Despite their sparsity, individual commands can exert sustained influence on locomotion behavior, which motivates our use of dual-channel drift modulation to capture both immediate and longer-term command effects.

## C. Extended Results

**Uncertainty Modeling via Sample Diversity** Table 5 and Table 6 investigate the ability of different models to represent predictive uncertainty under long-horizon, instruction-conditioned motion prediction. In addition to average FDE, which

measures pointwise prediction accuracy, we report a diversity metric computed as the mean pairwise distance among multiple stochastic rollouts. This metric characterizes the spread of the predictive distribution and reflects the model's capacity to represent multimodal futures. By jointly reporting accuracy and diversity at increasing prediction horizons, we aim to distinguish calibrated uncertainty modeling from uncontrolled noise amplification, particularly in long-horizon settings where intrinsic ambiguity becomes unavoidable.

*Table 5.* Long-horizon average FDE and diversity comparison on Babel.

| Method | $FDE_{avg}$ | | | | | | diversity | | | | | |
|---|---|---|---|---|---|---|---|---|---|---|---|---|
| | 10 | 20 | 30 | 40 | 50 | 60 | 10 | 20 | 30 | 40 | 50 | 60 |
| SLD-HMP | 5.037 | 6.514 | 7.126 | 7.600 | 7.318 | 7.521 | 7.004 | 9.156 | 10.523 | 11.554 | 11.288 | 11.624 |
| MoFlow | 0.018 | 0.039 | 0.062 | 0.084 | 0.104 | 0.123 | 0.014 | 0.032 | 0.053 | 0.073 | 0.092 | 0.107 |
| CogSDE | **0.010** | **0.025** | **0.042** | **0.058** | **0.073** | **0.086** | 0.006 | 0.018 | 0.033 | 0.047 | 0.060 | 0.070 |

*Table 6.* Long-horizon average FDE and diversity comparison on rat locomotion dataset.

| Method | $FDE_{avg}$ | | | | | | diversity | | | | | |
|---|---|---|---|---|---|---|---|---|---|---|---|---|
| | 10 | 20 | 30 | 40 | 50 | 60 | 10 | 20 | 30 | 40 | 50 | 60 |
| SLD-HMP | 3.67 | 7.55 | 11.40 | 14.92 | 18.01 | 20.74 | 1.54 | 3.03 | 4.56 | 5.84 | 7.13 | 8.13 |
| MoFlow | 3.64 | 9.48 | 16.69 | 24.17 | 30.84 | 35.91 | 2.49 | 7.23 | 13.05 | 18.71 | 23.32 | 26.73 |
| CogSDE | **1.58** | **4.64** | **9.16** | **14.39** | **19.64** | **24.27** | 1.02 | 3.57 | 7.90 | 13.38 | 19.09 | 24.11 |

On the BABEL human motion dataset, CogSDE models long-horizon uncertainty in a calibrated manner, maintaining low error while avoiding both mode collapse and excessive dispersion. Table 5 shows that CogSDE consistently achieves the lowest average FDE across all prediction horizons, indicating superior long-horizon accuracy compared to both SLD-HMP and MoFlow. At the same time, CogSDE exhibits non-trivial but controlled diversity that increases smoothly with the prediction horizon. In contrast, SLD-HMP exhibits unusually high diversity values together with substantially larger prediction errors, indicating that a subset of its stochastic samples fails to produce meaningful motion predictions. MoFlow yields relatively low errors but also exhibits limited diversity, indicating a tendency toward mode averaging. These results suggest that CogSDE achieves a more favorable accuracy–diversity trade-off, capturing meaningful uncertainty without sacrificing predictive precision on complex human motion data.

On rat locomotion data, CogSDE more effectively captures intrinsic behavioral uncertainty induced by sparse control commands, achieving both improved accuracy and principled uncertainty representation over long horizons. A similar but more pronounced trend is observed on the rat locomotion dataset in Table 6. As the prediction horizon increases, all methods experience growing average FDE and diversity, reflecting the inherently higher uncertainty in animal locomotion under external control. However, CogSDE consistently achieves substantially lower average FDE than the baselines across all horizons, while exhibiting diversity levels that scale appropriately with time. Notably, MoFlow shows rapidly increasing diversity accompanied by a sharp degradation in accuracy, whereas SLD-HMP maintains relatively low diversity but fails to capture the variability of future behaviors, resulting in higher long-horizon errors. These results indicate that CogSDE is better able to balance stochastic exploration and boundedness in biologically grounded motion prediction tasks.

**Qualitative Error Analysis on Long-Horizon Motion Prediction**    Qualitative analysis on representative BABEL sequences further illustrates how different models behave under long-horizon, instruction-conditioned motion prediction. Figure 9 presents four representative cases comparing ground truth (GT), our method (CogSDE), and SLD-HMP under identical historical observations and command conditions. These cases highlight distinct long-horizon failure patterns of latent-direction baselines, including stagnation of future dynamics, underestimation of motion amplitude, pose bias toward resting configurations, and over-extrapolation of future actions. In contrast, CogSDE maintains coherent temporal evolution across the prediction horizon, producing motion trajectories that better align with the intended commands while preserving plausible variability. Together with the quantitative results in Table 5, these visual examples suggest that explicitly controlled stochastic dynamics help mitigate common long-horizon degradation modes.

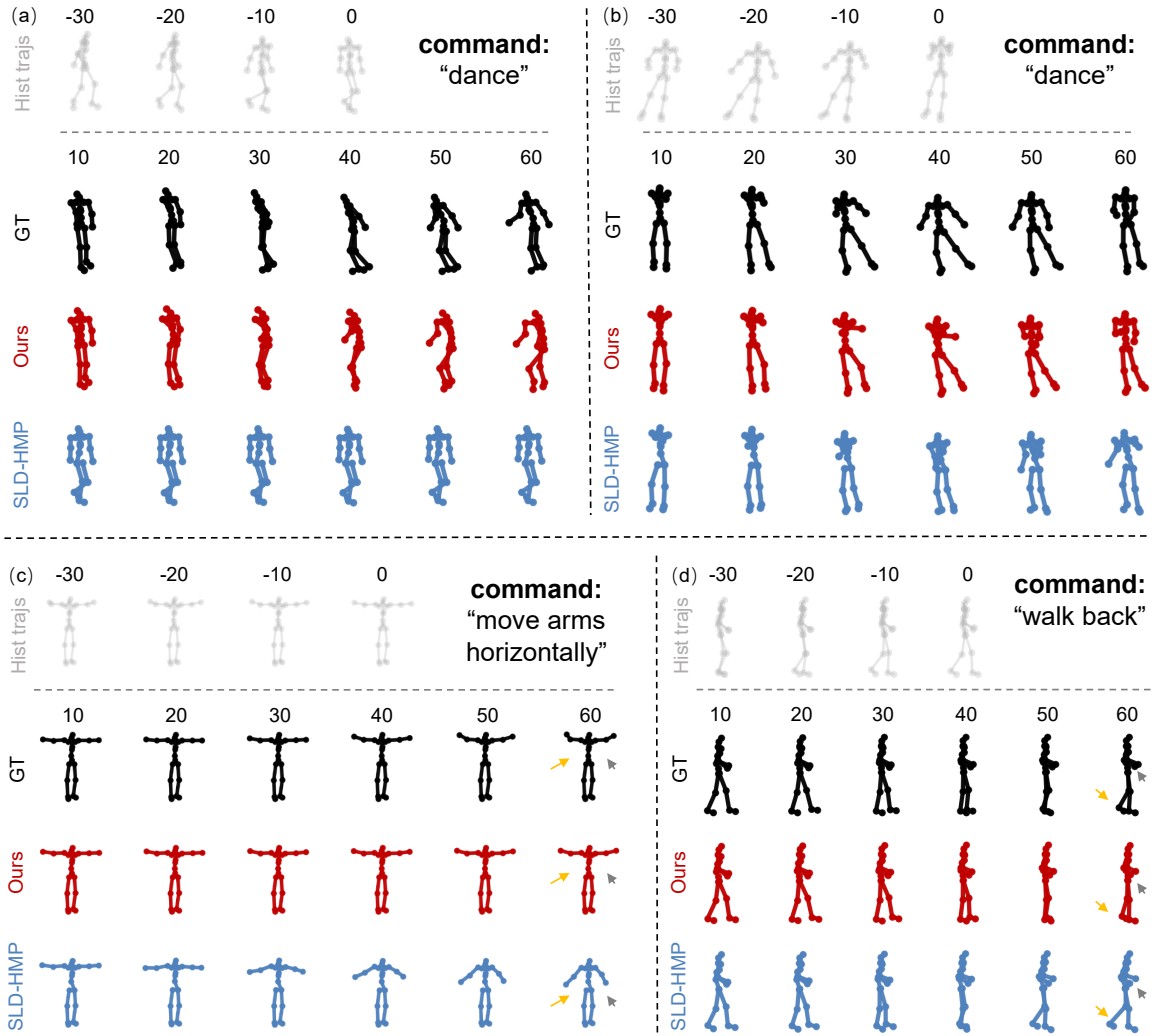

*Figure 9.* Qualitative comparison on the BABEL human motion dataset under long-horizon, instruction-conditioned prediction. For each case, we visualize the historical context (top, gray), followed by ground-truth future motion (GT, black), predictions from CogSDE (red), and SLD-HMP (blue). The same motion history and command sequence are provided to all methods, and future poses are shown at selected horizons (10–60 frames). (a) **Stagnation:** SLD-HMP produces near-static future poses under the "dance" command, failing to capture the evolving motion dynamics.(b) **Under-amplified motion:** SLD-HMP captures the coarse motion direction but underestimates the movement amplitude, particularly for leg trajectories, leading to noticeable discrepancies at the end of the horizon. (c) **Pose bias toward resting configurations:** For actions requiring sustained arm elevation ("move arms horizontally"), SLD-HMP exhibits a tendency to regress toward lower-arm poses, deviating from the ground truth (highlighted by arrows). (d) **Over-extrapolation:** Under the "walk back" command, SLD-HMP over-predicts future leg motion, producing exaggerated strides at 60 frames that are not supported by the ground truth. In contrast, CogSDE consistently preserves the intended motion trends across all cases, producing coherent long-horizon predictions that better align with the commanded behaviors.

**Comparison with action-conditioned latent dynamics models.** To further evaluate the necessity of the proposed controlled latent SDE formulation, we compare CogSDE with several representative latent dynamics models, including RSSM/Dreamer-style latent world models and latent autoregressive baselines implemented with GRU and Transformer dynamics. All methods share the same flow-based decoder, while differing only in the latent transition module.

As shown in Table 7, CogSDE consistently achieves the best long-horizon prediction performance on both the rat and BABEL datasets. In particular, on the rat dataset with a 60-frame prediction horizon, CogSDE reduces $ADE_{avg}$ from $16.47$ (RSSM) to $10.42$, substantially outperforming discrete-time latent autoregressive alternatives. Similar improvements are observed on the BABEL benchmark.

*Table 7.* Comparison with action-conditioned latent dynamics models and conditioning strategies.

| ID | Dataset | Encoder | Condition | ADE$_{min}$ ↓ | FDE$_{min}$ ↓ | ADE$_{avg}$ ↓ | FDE$_{avg}$ ↓ |
|----|---------|---------|-----------|------|------|------|------|
| 1 | rat | RSSM | concat+FiLM | 6.21 | 8.42 | 16.47 | 30.04 |
| 2 | rat | RSSM | attention | 6.20 | 8.14 | 16.04 | 29.55 |
| 3 | rat | Latent AR (GRU) | concat+FiLM | 9.42 | 13.53 | 17.63 | 33.97 |
| 4 | rat | Latent AR (Transformer) | concat+FiLM | 9.28 | 11.59 | 21.37 | 37.35 |
| 5 | rat | Latent AR (Transformer) | attention | 9.98 | 13.13 | 21.58 | 40.81 |
| 6 | rat | CogSDE (ours) | concat+FiLM | **3.79** | **6.54** | **10.42** | **24.27** |
| 7 | BABEL | RSSM | concat+FiLM | 0.022 | 0.035 | 0.044 | 0.089 |
| 8 | BABEL | Latent AR (GRU) | concat+FiLM | 0.060 | 0.105 | 0.123 | 0.323 |
| 9 | BABEL | Latent AR (Transformer) | concat+FiLM | 0.090 | 0.162 | 0.090 | 0.162 |
| 10 | BABEL | CogSDE (ours) | concat+FiLM | **0.018** | **0.032** | **0.042** | **0.086** |

*Table 8.* Comparison of prediction performance with MLP and flow-matching decoders.

| ID | Dataset | Encoder | Decoder | ADE$_{min}$ ↓ | FDE$_{min}$ ↓ | ADE$_{avg}$ ↓ | FDE$_{avg}$ ↓ |
|----|---------|---------|---------|------|------|------|------|
| 1 | rat | RSSM | MLP | 5.85 | 7.90 | 16.01 | 29.65 |
| 2 | rat | Latent AR (GRU) | MLP | 10.88 | 16.15 | 14.24 | 22.10 |
| 3 | rat | Latent AR (Transformer) | MLP | 8.02 | 10.13 | **9.04** | **17.18** |
| 4 | rat | CogSDE (ours) | MoFlow | **3.79** | **6.54** | 10.42 | 24.27 |
| 5 | BABEL | RSSM | MLP | 0.019 | **0.029** | **0.036** | **0.069** |
| 6 | BABEL | Latent AR (GRU) | MLP | 0.039 | 0.042 | 0.045 | 0.082 |
| 7 | BABEL | Latent AR (Transformer) | MLP | 0.046 | 0.060 | 0.051 | 0.091 |
| 8 | BABEL | CogSDE (ours) | MoFlow | **0.018** | 0.032 | 0.042 | 0.086 |

These results suggest that the advantage of CogSDE does not primarily arise from data efficiency or architectural scale, but from its ability to model partially observed and non-communicable latent decision processes under external interventions. Unlike discrete-time latent autoregressive models, the controlled latent SDE explicitly separates command-driven drift dynamics from stochastic diffusion, allowing the latent trajectory to evolve continuously under time-varying control inputs while preserving uncertainty over long horizons. This property is particularly important in cyborg-animal behavior prediction, where identical stimulation commands may induce variable behavioral responses due to unobserved cognitive states and internal decision variability.

**Necessity of the generative decoder.** To examine whether the flow-matching decoder is necessary under action-conditioned prediction, we replace the decoder with a simple MLP while keeping the latent backbone fixed. As shown in Table 8, MLP decoders achieve competitive results on BABEL, where human motion is smoother and the action-conditioned future is relatively less stochastic. However, their performance is less consistent on the rat dataset, where identical or similar commands can still induce variable responses due to unobserved internal states. In this setting, CogSDE with a flow-matching decoder achieves the best ADE$_{min}$ and FDE$_{min}$, indicating stronger ability to recover plausible best-case futures under behavioral uncertainty.

These results suggest that the flow-matching decoder is not mainly introduced for generic architectural complexity, but for modeling residual uncertainty that remains after conditioning on commands and latent dynamics. While a simple MLP can be sufficient when the conditional future is close to deterministic, the rat task retains substantial response variability even under command conditioning. The generative decoder therefore provides a more expressive conditional mapping from latent decision dynamics to future motion, complementing the controlled SDE backbone in stochastic animal behavior prediction.

**Sensitivity to the number of regimes.** We evaluate the sensitivity of CogSDE to the number of latent regimes $S$, which controls the granularity of the regime-based mixture in the command-conditioned drift and diffusion components. The default setting $S = 3$ is motivated by the coarse command-aligned behavioral modes in the rat task. As shown in Table 10, $S = 3$ achieves the best overall long-horizon prediction performance. Using fewer regimes slightly reduces flexibility, while increasing the number of regimes to $S = 4$ does not provide further gains and leads to larger average errors. These results suggest a practical trade-off between model expressiveness and stability, with $S = 3$ providing a balanced configuration.

*Table 9.* Cross-subject generalization under leave-one-rat-out evaluation.

| Train Rats | Test Rat | ADE$_{min}$ ↓ | FDE$_{min}$ ↓ | ADE$_{avg}$ ↓ | FDE$_{avg}$ ↓ |
|---|---|---|---|---|---|
| R1, R2 | R3 | 8.70 | 11.64 | 24.82 | 40.88 |
| R1, R3 | R2 | 6.02 | 6.50 | 19.11 | 30.45 |
| R2, R3 | R1 | 6.95 | 9.05 | 20.04 | 32.54 |

*Table 10.* Effect of the number of regimes on long-horizon prediction.

| Regimes | ADE$_{min}$ ↓ | FDE$_{min}$ ↓ | ADE$_{avg}$ ↓ | FDE$_{avg}$ ↓ |
|---|---|---|---|---|
| 2 | 3.84 | 7.29 | 11.27 | 24.86 |
| 3 | **3.79** | **6.54** | **10.42** | **24.28** |
| 4 | 3.88 | 7.41 | 11.89 | 27.15 |

**Cross-subject generalization.** We further evaluate cross-subject robustness using a leave-one-rat-out protocol, where the model is trained on two rats and tested on the held-out rat. As shown in Table 9, performance varies across held-out subjects, reflecting subject-specific differences in locomotion speed, response latency, and stimulation-induced behavior. Nevertheless, the model maintains stable prediction performance across all splits, suggesting that CogSDE captures shared latent structure beyond purely individual-specific patterns. These results also indicate that cross-subject prediction remains more challenging than within-subject evaluation, motivating future validation on larger animal cohorts and broader stimulation conditions.

**Predictive calibration analysis.** We further evaluate whether the stochastic rollouts of CogSDE provide meaningful predictive uncertainty. For each input history and command sequence, we draw multiple stochastic future predictions and use their sample variance as the predictive uncertainty. The variance is computed across sampled trajectories, keypoints, and coordinate dimensions, and is summarized either at each timestep or averaged over the full trajectory. Prediction error is computed from the mean prediction, using ADE for trajectory-level error and FDE for final-step error.

We then measure Pearson and Spearman correlations between predicted uncertainty and realized prediction error at both trajectory and timestep levels. As shown in Table 11, uncertainty is strongly correlated with error, with timestep-level correlations reaching 0.828 Pearson and 0.928 Spearman for ADE. This indicates that CogSDE tends to assign higher uncertainty to regions where prediction is more difficult, supporting the role of state-conditioned diffusion in modeling residual uncertainty under command-conditioned behavior prediction.

## D. Training and Implementation Details

This appendix provides implementation and training details to support reproducibility. Unless otherwise stated, all formulations follow the definitions introduced in the main paper.

**Training Overview.** In all experiments, we instantiate the flow-matching generator using MoFlow as a concrete implementation of conditional flow matching. This choice provides a stable and widely used backbone for supervising latent vector fields. Importantly, the proposed controlled SDE formulation operates at the level of latent vector-field parameterization and does not rely on MoFlow-specific architectural assumptions. MoFlow is therefore adopted solely as an implementation choice for reproducibility and comparison.

The training procedure, including latent initialization, controlled SDE rollout, and trajectory prediction supervision of the flow decoder, follows exactly the formulation described in Sections 3.1–3.2 of the main paper. In particular, external commands influence the prediction only through their modulation of the latent drift, and gradients propagate through the discretized SDE rollout over the full prediction horizon. All remaining architectural and optimization details are provided in the subsequent appendix sections.

**Objective Function.** We describe the exact training loss used in our implementation.

Given denoised predictions and ground-truth future trajectories, we compute a per-agent, per-timestep Euclidean error

$$e^{b,k,a,t} = \left\| x_{\mathbf{p}}^{b,k,a,t} - x_{\mathbf{f}}^{b,k,a,t} \right\|_2, \tag{26}$$

*Table 11.* Correlation between predicted uncertainty and prediction error.

| Granularity | Target | Pearson↑ | Spearman↑ | $p$-value |
|---|---|---|---|---|
| Trajectory | ADE | 0.646 | 0.553 | $< 10^{-50}$ |
| Trajectory | FDE | 0.682 | 0.560 | $< 10^{-50}$ |
| Timestep | ADE | 0.828 | 0.928 | $\approx 0$ |

where $b$ indexes the batch, $k \in \{1, \ldots, K\}$ indexes the sampled component, $a$ indexes agents, and $t$ indexes future timesteps. The error is aggregated over time by summation:

$$\bar{e}^{b,k,a} = \sum_t e^{b,k,a,t}. \tag{27}$$

A winner-take-all (best-of-$K$) selection is then performed independently for each agent:

$$k^{(b,a)} = \arg\min_k \bar{e}_{b,k,a}. \tag{28}$$

The per-sample regression loss is obtained by averaging the selected errors across agents:

$$\ell_b = \frac{1}{A} \sum_{a=1}^{A} \bar{e}_{b,k^{(b,a)},a}. \tag{29}$$

The final training objective is the batch average

$$\mathcal{L}_{\text{reg}} = \mathbb{E}_b[\ell_b]. \tag{30}$$

This loss term is the only quantity used for backpropagation in our experiments.

**Command Encoder.** External commands are provided as discrete events with sparse timing. To interface with the continuous-time controlled dynamics in Eq. (4), we encode commands into a time-varying continuous control signal $c_t \in \mathbb{R}^{d_u}$, which is consumed by the drift modulation modules in both the level and event channels.

For Rat-Locomotion dataset with sparse, step-wise commands and a small number of command types, we use a per-timestep 7-dimensional command vector consisting of: (i) a 4D one-hot category indicator over {none, forward, left, right}, (ii) a command strength scalar, (iii) a signed-strength scalar (left $< 0$, right $> 0$, others $= 0$), and (iv) a scalar $\Delta t_{\text{cmd}}$ measuring the number of steps since the last valid command.

For BABEL, commands do not include strength values and are specified as segment-level labels that remain valid for all frames within a time interval. We therefore encode commands using a compact 2-bit representation and expand it to a per-timestep sequence by assigning the same code to every frame in the corresponding interval. The resulting per-timestep command feature is a 13-dimensional vector.

We use a lightweight command encoder to map the command vector input into a continuous control embedding. Specifically, the categorical command indicator is mapped to a learned embedding vector, which is then concatenated with auxiliary command features when available. Here, $\Delta t_{\text{cmd}}$ denotes the number of discrete timesteps since the most recent valid command. Based on $\Delta t_{\text{cmd}}$, we construct continuous and scale-controlled time features, including an exponential decay term $\exp(-\Delta t_{\text{cmd}}/\tau)$ and a normalized log-time term $\log(1 + \Delta t_{\text{cmd}})/\log(1 + \text{time\_scale})$. These time features provide a bounded representation of command recency and are omitted when the underlying dataset does not provide step-wise command timing. The concatenated feature is projected by a single linear layer to produce $c_t$.

To ensure that "no-command" does not introduce an arbitrary category vector, the embedding for the "none" category is masked to zero, making differences such as $c_t - c_{t-1}$ meaningful.

**Latent SDE Discretization and Training Rollout.** The controlled SDE defined in Eq. (4) is numerically integrated using the Euler–Maruyama discretization scheme with a fixed step size $\Delta t = 0.033$, corresponding to a sampling rate of 30 Hz. The same discretization strategy and step size are used consistently during both training and inference. Latent dynamics are unrolled over the entire prediction horizon by iterating the discretized SDE forward in time.

*Table 12.* Consolidated architecture and training configuration for Rat-Loco and BABEL datasets.

| Category | Rat-Loco | BABEL |
|---|---|---|
| *Data and Prediction Setup* | | |
| Past frames $T_h$ | 30 | 30 |
| Future frames $T_f$ | 60 | 60 |
| Number of agents $A$ | 8 | 22 |
| Agent dimension | 2 | 3 |
| Time step $\Delta t$ | 0.033 (30 Hz) | 0.033 (30 Hz) |
| *Latent and Control Dimensions* | | |
| Latent dimension $d_z$ | 64 | 64 |
| Command dimension $d_u$ | 7 | 13 |
| Number of branches $K$ | 20 | 20 |
| *Context Encoder (MTREncoder)* | | |
| Input context channels | 6 | 9 |
| Model width | 128 | 128 |
| Attention layers | 4 | 4 |
| Attention heads | 8 | 8 |
| Neighbor agents | 8 | 22 |
| Attention dropout | 0.1 | 0.1 |
| *Motion Decoder (MTRDecoder)* | | |
| Decoder blocks | 4 | 4 |
| Model width | 128 | 128 |
| Attention heads | 8 | 8 |
| Attention dropout | 0.1 | 0.1 |
| *Optimization and Training* | | |
| Optimizer | AdamW | AdamW |
| Learning rate | $1 \times 10^{-3} \rightarrow 1 \times 10^{-6}$ | $1 \times 10^{-3} \rightarrow 1 \times 10^{-6}$ |
| Weight decay | 0.05 | 0.05 |
| Gradient clipping | 1.0 | 1.0 |
| Training epochs | 600 | 600 |
| Train batch size | 48 | 32 |
| Test batch size | 48 | 32 |
| Learning rate schedule | Cosine annealing + warmup | Cosine annealing + warmup |
| Random seed | 42 | 42 |

**Diffusion Parameterization.** The diffusion term $G_{\theta_g}(Z_t)$ follows the parameterization described in Section 3.3 of the main paper and is restricted to a diagonal form. In implementation, the diagonal diffusion is parameterized in log scale and allows state-dependent modulation. Specifically, diffusion scales are predicted per latent dimension and combined across regimes using the regime assignment weights defined in the controlled latent dynamics. The resulting effective diffusion is therefore a mixture of diagonal diffusion components. For numerical stability, the diffusion magnitude is bounded within a fixed range. No diffusion-specific regularization losses are used in the main training objective. An optional boundedness regularizer, introduced as a separate stability-oriented study, indirectly constrains the diffusion magnitude through the dissipativity condition.

**Architecture and Training Configuration.** Unless otherwise stated, all experiments are conducted using a fixed model architecture and a unified set of training hyperparameters. The overall framework consists of three main components: (i) a context encoder that processes historical observations and agent interactions, (ii) a controlled latent SDE that governs latent evolution under external commands, and (iii) a flow-matching-based decoder that generates future trajectories.

*Table 13.* Efficiency comparison on rat and BABEL datasets.

| Method | Dataset | Latency / Sample (ms) | Latency / Batch (ms) | Throughput (seq/s) | Mem (GB) |
|--------|---------|-----------------------|----------------------|--------------------|----------|
| TimeXer | rat | 0.48 | 1.93 | 2070.78 | 0.05 |
| DiffSTG | rat | 161.73 | 2587.69 | 6.18 | 0.89 |
| SLD-HMP | rat | 1.12 | 17.85 | 896.36 | 0.38 |
| MoFlow | rat | 6.94 | 110.98 | 144.17 | 0.11 |
| CogSDE | rat | 20.53 | 328.46 | 48.71 | 0.45 |
| TimeXer | BABEL | 0.16 | 2.49 | 6413.92 | 0.08 |
| DiffSTG | BABEL | 524.41 | 8390.62 | 1.91 | 2.63 |
| SLD-HMP | BABEL | 3.42 | 54.79 | 292.02 | 0.93 |
| MoFlow | BABEL | 3.90 | 1998.18 | 256.23 | 4.68 |
| CogSDE | BABEL | 42.50 | 339.97 | 23.53 | 0.59 |

**Runtime and resource comparison.** Table 13 reports inference latency, throughput, and GPU memory usage under a unified setting (batch size 16, stochastic sampling $K = 20$). CogSDE is slower than lightweight deterministic predictors, but remains competitive among stochastic generative models. In particular, it is substantially faster than diffusion-based methods while maintaining moderate GPU memory consumption.

