# OpenReview forum: "Controlled SDEs for Long-Horizon Motion Generation under Latent Decision Uncertainty"
_ICML.cc/2026/Conference — ICML 2026 regular_

### Official Review · Reviewer_NSeS · 2026-03-01

**Soundness:** 3
**Presentation:** 2
**Significance:** 2
**Originality:** 3
**Overall Recommendation:** 4
**Confidence:** 2

**Summary:**

The authors propose to use a hidden Markov model in the form of controlled SDE to be used as input augmentation in a generative model for motion prediction of, in particular, biological agents.

The controlled SDE is
$$
dZ_t = f_{\theta_f}(Z_t,u_t)dt + g_{\theta_g}(Z_t,u_t)dW_t
$$
aiming to capture long-time persistence of the control input $u_t$ present at time $t$ as well as the stochasticity of the effect.

The overall aim is to obtain a conditional distribution $p_\theta(x_{T_h+1:T_h+T_p}|x_{1:T_h},u_{1:T_h+T_p})$; it is not clear to this reviewer how the controlled SDE is incorporated in this.

The paper then evaluates the proposed methodology on two data-sets (rat locomotion, BABEL for human movement prediction) presenting results indicating that the proposed methodology outperforms benchmark methods. Finally, an ablation study is provided which strongly indicates the need for the complexity of introducing the hidden state as a controlled SDE.

**Compliance With Llm Reviewing Policy:**

Affirmed.

**Final Justification:**

The authors addressed the concerns raised in the review and the score has been adjusted accordingly (weak recejct -> weak accept).

This reviewer isn't really sufficiently familiar with the techniques and state of art in the area so the AC should give more weight to the recommendations of other reviewers.

**Key Questions For Authors:**

Q1 Would it be possible to use appropriate logical quantifiers for all the variables in Assumption 3.1? I.e. $\exists \alpha > 0, \beta \geq 0$ such that for all $\ldots$.

Q2 This relates to W1. The reviewer understands that modelling the conditional density (1) is in general the aim in motion prediction in presence of external control. Where and how is this later augmented to include the latent variable? Or perhaps it isn’t?

Q3 This relates to W2: Why not do the analysis for the discrete explicit Euler scheme which is use in implementation?

Q4: This relates to W2: Why did you choose to not enforce the constraint (7) in implementation?

**Limitations:**

It is not clear how this methodology would perform on other data sets.

The data sets used seem relatively low-dimensional in both control and observation space dimensions - it is not clear how well this would scale with dimensions.

Given the already mentioned W1 and W3 the potential for reproducibility of the results is poor.

Note that below this reviewer gives themselves only "2" on confidence. However that should be read as "1" for the motion prediction, as the reviewer knows nothing about this area. However SDEs, stability, numerics for SDEs are the reviewer's bread and butter and the confidence there is "5". Neverthless, this is mainly an applied paper and should be judged on the merits of the application.

**Strengths And Weaknesses:**

### Strengths

S1 The presented benchmark results seem to indicate good performance of the proposed method.

S2 The propose methodology of modelling a latent state using an SDE seems methodologically sound and original within the application domain (though the reviewer isn’t an expert on locomotion prediction modelling).

### Weaknesses

W1 Despite Figure 2 this reviewer could not really see how the entire prediction system is constructed and trained. Could the authors provide detail on how the latent state is used to construct the conditional distribution $p_\theta(x_{T_h+1:T_h+T_p}|x_{1:T_h},u_{1:T_h+T_p})$?

W2 (a) Theorem 3.1, while technically correct, does not apply to the setting of the paper because of course the algorithm uses an explicit Euler stepping scheme (as stated in Appendix D.4). Of course Lyapunov stability can be proved under the present coercivity condition, however some kind of smoothness and a step size restriction would be required. Moreover, (b), since Assumption 3.1 is not enforced during training it’s not clear what the result of Theorem 3.1 would bring. Instead, the authors could just use three of the four panels in Figure 6 to conclude stability empirically. Finally, (c), Theorem 3.1 is trivial for anyone who’s familiar with Ito’s formula. Thus, in the opinion of this reviewer, it looks like an attempt to inject some “theory” into what is an empirical paper (and hopefully a good empirical paper). The space would be better employed to address W1 so that non-experts in locomotion prediction can appreciate how the machinery is set up.

W3 The source code doesn’t seem to be available and thus there is no way to even partially (if not all data sets are public) reproduce the results. The authors should aim to provide a synthetic data set that demonstrated the approach end-to-end.

---

> ### Author Rebuttal · Authors · 2026-03-31
>
> We thank the reviewer for the constructive feedback and for recognizing the soundness of our SDE-based latent modeling and its promising empirical performance (S1–S2). Below we clarify the main concerns on formulation, theory, and reproducibility.
>
> ---
> **Q1 Rewrite Assumption 3.1**: We thank the reviewer for pointing out this lack of precision. Here is the revised version:
>
> **Assumption 3.1 (Uniform Dissipativity)**:
> There exist constants $\alpha > 0$ and $\beta \ge 0$ such that for all latent states $z \in \mathbb{R}^{d_z}$ and bounded control inputs satisfying $ ||u_t|| \leq U_{\text{max}}$, the drift and diffusion functions satisfy
>
> $2\langle z, f_\theta(z,u)\rangle + \|G_\theta(z)\|_F^2
> \le -2\alpha \|z\|^2 + \beta.$
>
> ---
> **W1 & Q2: Implementation details.** We thank the reviewer for the question. We clarify the probabilistic formulation, latent dynamics, and training pipeline as Fig. 1 (https://anonymous.4open.science/r/adfhoaqhif/fig1.jpg).
>
> **(1) Latent stochastic formulation**
>
> We reformulate the objective via a latent trajectory $Z_{\text{fut}}:=Z_{T_h:T_h+T_p}$:
>
> $p_\theta(x_{\text{fut}}\mid x_{\text{hist}},u)
> =\int p_\theta(x_{\text{fut}}\mid Z_{\text{fut}})p_\theta(Z\mid x_{\text{hist}}, u)dZ.$
>
> Instead of a static distribution, $p_\theta(Z\mid x_{\text{hist}},u)$ is induced by a **controlled SDE**. The process is initialized by $Z_{T_h}=\phi_\theta(x_{\text{hist}},u_{\text{hist}}),$
> and evolves under time-varying commands, yielding a continuous-time latent decision trajectory as stated in the main text.
>
> **(2) Conditioning via latent modulation**
>
> The decoder is conditioned on a fused token:
> $c=\mathrm{MLP}([y_{\mathrm{emb}}, t_{\mathrm{emb}}, \text{context}, z_0]),$
> together with the latent state $Z_t$ at step $t$.
>
> We compute the modulated condition via FiLM:
> $
> \tilde{c}_t = \mathrm{FiLM}(c, Z_t)
> = \Gamma(Z_t) \odot c + B(Z_t),
> $
> where $\Gamma(\cdot), B(\cdot): \mathbb{R}^{d_z} \to \mathbb{R}^{d_c}$ are learnable mappings,
> and $\odot$ denotes element-wise multiplication.
>
> The flow model uses $\tilde c$ as condition:
> $
> v_\theta(y_t,t\mid \tilde c),\quad
> y_{t+\Delta t}=y_t+v_\theta\Delta t.
> $
> When $t=1$, the final state $y_1$ corresponds to the predicted trajectory $x_{\mathrm{fut}}$.
>
> **(3) Training objective**
>
> We train end-to-end with a **trajectory-level regression objective** $\mathcal{L_{\text{reg}}}$ (defined in response to Reviewer wfsF, W2), and **Boundedness regularization** $\mathcal{L_{\text{bnd}}}$ (defined in response to Reviewer RZtT, W1, W6, Q1) to stabilize latent SDE dynamics.
>
> ---
> **W2,Q3,Q4: Theoretical rigor and effectiveness.** We thank the reviewer for the comments. We address the concerns by separating the issues below.
>
> **(1) Theory–implementation mismatch (Q3)**
>
> We agree that the implementation relies on Euler–Maruyama discretization, and a fully discrete-time analysis would be precise. In practice, we verify that the chosen step size yields stable rollouts: both the latent energy $|Z_t|^2$ and the dissipativity-related quantity remain bounded throughout prediction. This suggests that the discretized dynamics are consistent with the continuous-time stability intuition. A rigorous discrete-time analysis would require additional assumptions and is left as future work.
>
> **(2) Theory not used in training (Q4)**
>
> We agree that, in the original version, constraints was not enforced during training.
>
> In the revised version, we incorporate the dissipativity condition as a **soft constraint $\mathcal{L}_{\text{bnd}}$** in the training objective with a weight $\lambda_{\mathrm{bnd}}$. The results are shown in Table 1. Empirically, this leads to:
>
> 1. Moderate regularization ($\lambda_\mathrm{bnd}=0.001–0.003$) improves accuracy (ADE_min, FDE_min), indicating more precise trajectory samples.
> 2. Larger $\lambda_\mathrm{bnd}$ increases ADE_avg and FDE_avg, suggesting a trade-off between stability and sample diversity.
>
> **Table 1.** Effect of $\lambda_{\mathrm{bnd}}$ on prediction performance.
> |$\lambda_{\mathrm{bnd}}$|ADE_min|FDE_min|ADE_avg|FDE_avg|
> |-|-|-|-|-|
> |0|3.79|6.54|10.42|24.27|
> |0.001|3.65|6.31|12.37|27.90|
> |0.003|3.67|6.37|11.48|24.99|
>
> **(3) Role and value of the theory**
>
> We agree that the theoretical result is standard and not a primary contribution. Its role is to provide a **minimal, principled justification** for the latent dynamics, offering (i) interpretability of control–uncertainty interactions, (ii) guidance for drift/diffusion design, and (iii) a basis for empirical diagnostics and regularization (Fig. 6, $L_{\text{bnd}}$). We therefore position it as **supporting analysis** for an otherwise empirical study.
>
> ---
> **W3: Reproducibility.** We thank the reviewer for this comment. The code and rat skeleton dataset are included in the supplementary ZIP, enabling full reproducibility. If permitted by the review policy, we will additionally provide an anonymized repository before the rebuttal deadline.

---

> > ### Author Rebuttal · Reviewer_NSeS · 2026-04-01
> >
> > The concerns of this (non-expert) reviewer have been mostly resolved and this reviewer will update the score to "weak accept" based on the responses provided.
> >
> > The reviewer hopes that the comments made will be used to improve the next version of the manuscript.

---

> > > ### Author Response · Authors · 2026-04-02
> > >
> > > Thank you for your thoughtful review and for updating your assessment. We appreciate your recognition that the responses have addressed the main concerns. We will carefully incorporate your comments to further improve the clarity and quality of the manuscript in the next revision.

---

### Official Review · Reviewer_RZtT · 2026-03-11

**Soundness:** 3
**Presentation:** 3
**Significance:** 3
**Originality:** 4
**Overall Recommendation:** 4
**Confidence:** 3

**Summary:**

This paper proposes CogSDE, a controlled stochastic differential equation framework for long-horizon motion prediction under external commands. The key perspective is that future motion is governed by latent decision uncertainty, where unobserved internal decision states evolve stochastically over time. To model this process, the paper introduces a controlled SDE for latent dynamics: the drift is modulated by a dual-channel control mechanism, with one channel capturing sustained command effects and the other capturing transient changes; the diffusion is state-dependent and diagonal, modeling intrinsic stochasticity in the latent decision process. The paper also provides a mean-square boundedness analysis under a one-sided dissipativity assumption. Experiments on a command-driven rat locomotion dataset and the BABEL dataset show improved long-horizon prediction accuracy and better distribution-level generation quality compared with several baselines.

**Compliance With Llm Reviewing Policy:**

Affirmed.

**Final Justification:**

The paper presents a novel controlled SDE framework for long-horizon motion prediction with commands, with a well-motivated design and clear originality. The dual-channel control and stochastic latent dynamics are technically sound, and experiments show consistent gains on long-horizon metrics.

The rebuttal addresses several key concerns. In particular, the added dissipativity regularization improves the connection between theory and training, and the additional efficiency and ablation results strengthen the empirical evaluation. However, some limitations remain. Cross-subject generalization is still limited, and baseline comparisons are not yet fully exhaustive, which weakens the overall empirical strength.

Overall, I find the work technically solid and original, with meaningful contributions, though the evaluation is somewhat limited. The rebuttal strengthens my confidence but does not change my overall assessment. I therefore maintain a Weak Accept recommendation.

**Key Questions For Authors:**

1. Did you try turning the dissipativity-related condition into an explicit regularizer or constraint during training? If so, how did it affect prediction accuracy and stability?

2. How sensitive is the method to the number of regimes and other structural hyperparameters? Is there a clear trade-off between performance and stability?

3. What is the practical training/inference cost of CogSDE for long-horizon rollout? Could you report runtime or resource comparisons against SLD-HMP, MoFlow, or TimeXer?

4. Why were the generative baselines not compared under stronger command-aware settings? Could future versions include more competitive conditional generative baselines?

5. The rat dataset has limited numbers of independent animals and trials. Have you analyzed cross-subject generalization or robustness under different stimulation strengths and response delays?

**Limitations:**

The theory supports interpretability but not deployment-level safety, especially in BCI or intervention settings where long-horizon errors may accumulate. In addition, the experiments are still limited in scale and do not fully establish robustness across subjects, environments, or more complex control settings. A clearer discussion of these limitations and possible failure modes would strengthen the paper.

**Strengths And Weaknesses:**

**Strengths:**

1. The paper studies long-horizon motion prediction under explicit commands, especially in biologically grounded and BCI-related settings, where latent decision uncertainty is central.

2.  A controlled SDE is a natural framework for jointly modeling continuous-time dynamics, command interventions, and intrinsic uncertainty. The dual-channel drift design is also well aligned with persistent versus abrupt command effects.

3. Although the boundedness result is assumption-based, it provides a useful explanation for long-horizon stability.

4. CogSDE achieves the best overall long-horizon performance on both Rat Locomotion and BABEL in terms of min-ADE and avg-ADE, suggesting gains in both accuracy and distribution control.

5. Drift modulation and stochastic diffusion each contribute benefits, while the full model is most stable on distribution-level metrics, supporting the claimed complementarity.

**Weaknesses:**

1. The paper explicitly states that dissipativity is not enforced during training, so the analysis is better viewed as a post-hoc explanation rather than a strict safety guarantee.

2. Some generative baselines are evaluated in their original unconditioned form rather than under equally expressive command-aware modeling, which weakens the comparison for instruction-driven generation.

3. The paper does not report training or inference time, latency, FLOPs, or computational comparisons with baselines, which matters for long-horizon rollout settings.

4. For example, there is no sensitivity study for structural choices such as the number of regimes.

5. The rat dataset is distinctive and interesting, but the number of independent animals and trials is still relatively small, so broader generalization remains less convincing.

6. The paper provides post-hoc empirical diagnostics for stability, but does not yet show how the theoretical conditions could be turned into trainable or verifiable constraints.

---

> ### Author Rebuttal · Authors · 2026-03-31
>
> Thank you for the positive evaluation. The core idea is well received. The concerns on theory–training alignment, baseline fairness, and experimental completeness are addressed with additional experiments to strengthen the paper.
>
> ---
> **W1, W6, Q1: Dissipative constraints.** We thank the reviewer for this important suggestion. We incorporate the dissipativity condition into training as an explicit regularizer, implemented as a soft penalty applied directly to the latent SDE rollout. Specifically, at each step $k$, we compute
>
> $r_k = 2\langle z_k, f_\theta(z_k,u_k)\rangle + |G_\theta(z_k)|_F^2 + 2\alpha |z_k|^2,$
>
> and penalize violations $v_k = r_k - \beta$ using a temperature-controlled softplus:
>
> $\mathcal L_{\mathrm{bnd}} = \frac{\sum_k w_k \tau \mathrm{softplus}(v_k / \tau)}{\sum_k w_k}.$
>
> Here, $w_k$ emphasizes later rollout steps, and $|G_\theta|\_F^2$ reduces to the sum of squared diagonal diffusion scales. The constraint is enforced on latent dynamics, consistent with the theory. We adopt a warmup + ramp schedule for $\lambda_{\mathrm{bnd}}$ to avoid interfering with the main objective.
>
> **Table 1.** Effect of the dissipativity regularization weight $\lambda_{\mathrm{bnd}}$ on prediction performance.
> |$\lambda_{\mathrm{bnd}}$|ADE_min|FDE_min|ADE_avg|FDE_avg|
> |-|-|-|-|-|
> |0|3.79|6.54|10.42|24.27|
> |0.001|3.65|6.31|12.37|27.90|
> |0.003|3.67|6.37|11.48|24.99|
>
> This small weight acts as a gentle bias. Table 1 shows that the dissipativity term effectively reduces violations without over-constraining. A weight sweep confirms that moderate values improve stability, while larger ones may harm accuracy.
>
> ---
> **W2& Q4: Command-aware baselines.** We acknowledge that the comparison is not fully exhaustive. Several baselines are already command-aware (e.g., SLD-HMP, MoFlow), while only DiffSTG lacks explicit conditioning and performs significantly worse. We agree that including stronger conditional generative baselines would improve fairness.
>
> Motivated by this suggestion, we compare different conditioning strategies across encoders and fusion mechanisms (see response to U9Zm, W4 & Q2 for details). The results show that improved conditioning alone yields limited gains, indicating that the main advantage comes from modeling stochastic latent dynamics rather than the conditioning interface.
>
> ---
> **W3 & Q3: Resource comparisons.** We thank the reviewer for this important point. Runtime and resource comparisons are reported under a unified setting (batch size = 16, multimodal sampling K = 20, 20 FM steps).
>
> As shown in Table 2, CogSDE is slower than lightweight deterministic baselines (TimeXer), but remains competitive among generative models. It is faster than diffusion-based methods (DiffSTG) and more memory-efficient than flow-based models (MoFlow), while maintaining moderate runtime (20.53 ms/sample on rat, 42.50 ms/sample on BABEL). Overall, CogSDE offers a favorable efficiency–accuracy trade-off, with clear gains in distributional metrics (avg-ADE, diversity) central to our task.
>
> **Table 2.** Efficiency comparison on rat and BABEL datasets.
>
> |Method|Dataset|Latency/Sample(ms)|Latency/Batch(ms)|Throughput(seq/s)|Mem(GB)|
> |-|-|-|-|-|-|
> |Diffstg|rat|161.73|2587.69|6.18|0.89|
> |SLD-HMP|rat|1.12|17.85|896.36|0.38|
> |Timexer|rat|0.48|1.93|2070.78|0.05|
> |MoFlow|rat|6.94|110.98|144.17|0.11|
> |CogSDE|rat|20.53|328.46|48.71|0.45|
> |SLD-HMP|babel|3.42|54.79|292.02|0.93|
> |Timexer|babel|0.16|2.49|6413.92|0.08|
> |Diffstg|babel|524.41|8390.62|1.91|2.63|
> |MoFlow|babel|3.90|1998.18|256.23|4.68|
> |CogSDE|babel|42.50|339.97|23.53|0.59|
>
> ---
> **W4 & Q2: Regimes number.** The default choice (M=3) follows coarse command-aligned modes (forward/left/right). We will include a regime-number ablation (M=2,3,4). Preliminary results (Table 3) show M=3 captures main behaviors, while fewer regimes reduce flexibility and more regimes yield limited gains with higher cost.
>
> **Table 3**. Effect of the number of regimes on long-horizon prediction.
> |regimes|ADE_min|FDE_min|ADE_avg|FDE_avg|
> |-|-|-|-|-|
> |2|3.84|7.29|11.27|24.86|
> |3|3.79|6.54|10.42|24.28|
> |4|3.88|7.41|11.89|27.15|
>
> ---
> **W5&Q5: Cross-subject generalization.** We thank the reviewer for this observation. While the number of animals is limited in controlled cyborg experiments, our goal is to model shared latent dynamics under partial observability rather than individual-specific patterns.
>
> **Table 4**. Cross-subject generalization under leave-one-rat-out evaluation.
> |Train Rats|Test Rat|ADE_min|FDE_min|ADE_avg|FDE_avg|
> |-|-|-|-|-|-|
> |R1,R2|R3|8.70|11.64|24.82|40.88|
> |R1,R3|R2|6.02|6.50|19.11|30.45|
> |R2,R3|R1|6.95|9.05|20.04|32.54|
>
> Under leave-one-rat-out evaluation (Table 4), performance degrades due to subject-specific dynamics such as speed and delay, but remains stable. This variability reflects the inherent difficulty of cross-subject prediction, while suggesting the model captures shared structure and adapts to individual differences, consistent with the latent dynamics formulation.

---

> > ### Author Rebuttal · Reviewer_RZtT · 2026-04-01
> >
> > The rebuttal addresses several of my original concerns with concrete additional experiments. In particular, the dissipativity regularization results reduce the gap between the theoretical stability argument and the actual training procedure, and the added efficiency comparison and regime-number ablation improve the empirical picture.
> >
> > However, the new results also make the remaining limitation clearer rather than removing it. The leave-one-rat-out evaluation shows a substantial performance drop, which suggests that the current model still depends heavily on subject-specific dynamics and that cross-subject generalization remains limited. The discussion of command-aware baselines is improved, but the comparison is still not fully exhaustive, so the empirical case is stronger than before but not yet fully definitive.
> >
> > Overall, the rebuttal strengthens my original technical assessment, but it does not change my bottom-line view. I remain at Weak Accept: the paper has a real technical contribution, but its generalization bottleneck is still an important open issue.

---

> > > ### Author Response · Authors · 2026-04-02
> > >
> > > Thank you for your careful evaluation and for acknowledging the improvements brought by the additional experiments, particularly on dissipativity regularization, efficiency, and regime ablation.
> > >
> > > We also appreciate your insight on the cross-subject generalization limitation; we agree that the leave-one-rat-out results highlight a remaining gap, and we will explicitly clarify this limitation and position it as an important direction for future work.
> > >
> > > We are grateful for your balanced assessment and recognition of the paper’s technical contribution.

---

### Official Review · Reviewer_U9Zm · 2026-03-13

**Soundness:** 3
**Presentation:** 3
**Significance:** 2
**Originality:** 2
**Overall Recommendation:** 5
**Confidence:** 4

**Summary:**

This paper proposes CogSDE, a controlled SDE framework for long-horizon, command-driven motion prediction. Unlike prior work, which typically assumes static commands, this framework predicts motion under continuous, time-varying commands, while modeling stochasticity intrinsically.

Concretely, the CogSDE framework consists of (1) a latent-space SDE to model the dynamics and (2) a flow-matching decoder to map the latent states to outputs. The drift term of the latent SDE is modulated by continuous commands via two channels: sustained control $c_t$ (function of current timestep control) and transient event $c_t'$ (delta between current and previous sustained control). To capture stochasticity, the noise scale of the diffusion term is conditioned on latent state. The parameters of the SDE are trained end-to-end through discretized rollout over the full prediction horizon.

The paper evaluates CogSDE on (1) a custom rat locomotion dataset and (2) a human pose dataset (BABEL). Baselines include transformer-based deterministic predictors (TimeXer), diffusion-based unconditional predictors (DiffSTG, MoFlow), and command-aware generative models (SLD-HMP). CogSDE achieves the lowest prediction error of all the methods across both benchmarks. Each design choice of CogSDE is validated with ablation experiments.

Last but not least, the paper conducts a mean-squared boundedness analysis, deriving a sufficient condition for the boundedness of latent stochastic dynamics.

**Compliance With Llm Reviewing Policy:**

Affirmed.

**Final Justification:**

The rebuttals have addressed my main concern, which is the lack of comparison to autoregressive diffusion models. This changed my evaluation. Overall, the paper's claims are backed up with strong empirical and theoretical results, and I recommend acceptance.

**Key Questions For Authors:**

1. Can you directly learn the SDE in the input space? For image generation, a latent space is required because of the high dimensionality of the input. However, the motion space is not as high-dimensional.
2. Can you compare to a baseline that uses autoregressive latents + flow matching decoder, conditioning on actions via cross-attention? This is a standard approach for video generation. An even stronger baseline is diffusion-forcing / self-forcing.
3. Does the proposed method scale to video generation?

Typos:
1. Line 52 right: a unified
2. Figure 2: output box should be $x_{T_{h} + 1}$ instead of $x_{T_{h+1}}$. Upper middle box: $Z_{T_{f}} \rightarrow Z_{T_{h} + T_{p}}$.

**Limitations:**

The authors do not address the limitations of their work. The limitations are scalability to high-dimensional spaces.

**Strengths And Weaknesses:**

**Strengths**
1. The problem formulation and proposed method are sound.
2. The proposed method is supported by solid empirical and theoretical results
  a. Empirically, CogSDE outperforms baselines across two evaluation suites.
  b. Theoretically, the second moment of the SDE is bounded (Theorem 3.1) assuming one-sided dissipativity (Assumption 3.1). This is confirmed by targeted experiments in Section 4.4 (Figure 6).
  c. The individual design choices are supplemented by ablations in Section 4.3

**Weaknesses**
1. The contributions lack novelty. Continuous commands and stochasticity are well-studied in the literature. In robotics, for example, numerous studies in the literature have examined the problem of uncertainty-aware action-conditioned world modeling [1, 2, 3].
2. The proposed method features strong inducive bias (e.g., persistent and transition modulation to the diffusion term). It is unclear how these generalize to broader domains.
3. The empirical experiments are low-dimensional, and it's unclear how well the method scales to high-dimensional spaces.
4. The experiments did not compare to strong baselines such as autoregressive latents + diffusion decoder.

[1] Yanjiang Guo, Lucy Xiaoyang Shi, Jianyu Chen, Chelsea Finn. Ctrl-World: A Controllable Generative World Model for Robot Manipulation. ICLR 2026.

[2] Julian Quevedo, Ansh Kumar Sharma, Yixiang Sun, Varad Suryavanshi, Percy Liang, Sherry Yang. WorldGym: World Model as An Environment for Policy Evaluation. ArXiv 2025.

[3] Chenhao Li, Andreas Krause, Marco Hutter. Robotic World Model: A Neural Network Simulator for Robust Policy Optimization in Robotics. ArXiv 2025.

---

> ### Author Rebuttal · Authors · 2026-03-31
>
> We thank the reviewer for the constructive feedback. We appreciate the positive assessment of our problem formulation, empirical validation, and theoretical analysis. To address the concerns, we clarify the novelty and positioning of our approach, elaborate on the role of inductive bias, discuss scalability considerations, and augment our evaluation with additional baselines.
>
> ---
> **W1: Novelty.** We agree that action-conditioned stochastic world models have been studied. However, our work targets a distinct and underexplored region:
>
> > **Latent decision uncertainty**, where the internal dicision states governing behavior are partially observable and inherently stochastic, poses a fundamental challenge for long-horizon motion prediction under external command modulation.
>
> This differs from prior settings: in robotics, agents typically have known policy-to-action mappings; in human motion, intent is externally specified and largely aligned with given instructions.
>
> Our setting instead involves (i) stochastic latent decision dynamics, (ii) partial alignment between external commands and intrinsic behavior, and (iii) error accumulation over long horizons. We address these challenges via a latent SDE for state-dependent uncertainty, a structured control decomposition for intervention–intrinsic coupling, and a dissipativity-based formulation for stable long-horizon rollouts.
>
> ---
> **W2: Generalization ability.** We thank the reviewer for this important point. We clarify that the proposed modulation is applied to the **drift** rather than the diffusion term; diffusion remains **state-dependent** and models uncertainty.
>
> We acknowledge that our design introduces inductive bias, but it is not restrictive: it reduces to a standard latent SDE when modulation is removed, reflects a general temporal pattern of persistent vs. transient effects, and enables modeling of hidden decision uncertainty and stable long-horizon prediction.
>
> ---
> **W3 & Q3: High-dimensional expansion.** We thank the reviewer for raising the question of scalability. We address W3 and Q3 jointly, as video generation represents an extreme high-dimensional setting.
>
> 1. We are interested in extending to higher-dimensional domains (e.g., video), but such evaluations require substantially more computation and are beyond the current scope; we leave this as future work.
> 2. Our method operates in a latent dynamical space and is not tied to a specific modality; with suitable encoders/decoders (e.g., latent video representations), the same formulation can be applied.
> 3. We are also interested in biologically grounded extensions (e.g., neural signals and 3D musculoskeletal models), which we will discuss in the limitations and future work.
> ---
> **W4 & Q2: Stronger baselines.** We thank the reviewer for this suggestion. We have added stronger baselines following the proposed paradigm as shown in Table 1. In particular, latent AR + generative decoder and attention-based conditioning reflect widely used pipelines where actions are injected as external inputs. All methods are evaluated under the same protocol (K=20 sampling, min/avg ADE/FDE), ensuring fair comparison.
>
> **Table 1**. Comparison of autoregressive, RSSM, and our method, along with different conditioning strategies.
> |id|dataset|Encoder|Condition|ADE_min|FDE_min|ADE_avg|FDE_avg|
> |-|-|-|-|-|-|-|-|
> |1|rat|RSSM|concat+FiLM|6.21|8.42|16.47|30.04|
> |2|rat|RSSM|attention| 6.20 | 8.14 | 16.04 | 29.55 |
> |3|rat|Latent AR|concat+FiLM|9.28|11.59|21.37|37.35|
> |4|rat|Latent AR|attention|9.98 | 13.13 | 21.58 | 40.81 | 25.35 |
> |5|rat|CogSDE|concat+FiLM|**3.79**|**6.54**|**10.42**|**24.27**|
> |6|babel|RSSM|concat+FiLM|0.022|0.035|0.044|0.089|
> |8|babel|Latent AR|concat+FiLM|0.090|0.162|0.090|0.162|
> |10|babel|CogSDE|concat+FiLM|**0.018**|**0.032**|**0.042**|**0.086**|
>
> While RSSM and latent AR benefit from stochastic decoding, they still exhibit higher avg-ADE/FDE and a weaker diversity–accuracy trade-off. Cross-attention treats actions as external inputs without modeling their interaction with the dynamics, leading to error accumulation and poorly calibrated uncertainty.
>
> ---
> **Q1: Necessity of the latent space.** We agree that learning the SDE directly in the input space is feasible given the relatively low dimensionality of motion data. However, our use of a latent space is primarily motivated by partial observability: the decision states governing future behavior are hidden and stochastic, and cannot be directly inferred from observed keypoints. An input-space SDE would mix observable motion with latent decision dynamics, whereas our formulation explicitly separates the two.
>
> ---
> **Typo:** We thank the reviewer for noting these typos. We will correct them in the revision (Line 52 wording, Figure 2 output $x_{T_h+1}$, and latent transition notation).

---

> > ### Author Rebuttal · Reviewer_U9Zm · 2026-04-03
> >
> > Thanks for the clarifications and additional experiments. I am surprised by the latent autoregressive model results. Can you describe the model size of each method in the table?
> >
> > > Cross-attention treats actions as external inputs without modeling their interaction with the dynamics, leading to error accumulation and poorly calibrated uncertainty.
> >
> > Can you elaborate on this statement? Cross attention modulates the attention weights of the autoregressive transform and interacts with the dynamics.
> >
> > > An input-space SDE would mix observable motion with latent decision dynamics, whereas our formulation explicitly separates the two.
> >
> > Why is explicit separation of observable motion and decision dynamics better than end-to-end modeling?

---

> > > ### Author Response · Authors · 2026-04-03
> > >
> > > We thank the reviewer for these insightful follow-up questions. They help clarify the fairness of the comparison, the role of cross-attention conditioning, and our motivation for explicitly separating latent decision dynamics from observable motion.
> > >
> > >
> > > ---
> > > **Q1: Model size and fairness of comparison.**
> > >
> > > We agree that the latent autoregressive results require careful evaluation, and we report parameter counts in Table 1. All models have comparable sizes (≈3.6M–4.1M). More importantly, they share the same pipeline: the same MTR encoder, conditioning pathway, and flow-matching decoder under identical training/evaluation settings. The primary difference lies in the latent dynamics module and how control is incorporated. This indicates that the performance gain of CogSDE is not due to model capacity.
> > >
> > > **Table 1. Parameter Counts. (All with the same flow-matching decoder.)**
> > > |dataset| Model | Total Params | Trainable Params |
> > > | --- |--- | ---: | ---: |
> > > | rat | RSSM + concat | 3,648,130 | 3,648,130 |
> > > | rat | RSSM + attention | 3,956,361 | 3,956,361 |
> > > | rat | latent AR (transformer) + concat | 3,796,226 | 3,796,226 |
> > > | rat | latent AR (transformer) + attention | 4,096,265 | 4,096,265 |
> > > | rat | CogSDE **(ours)** | 3,969,576 | 3,969,576 |
> > > | babel | RSSM + concat  | 3,663,619 | 3,663,619 |
> > > | babel | latent AR (transformer) + concat | 3,810,179 | 3,810,179 |
> > > | babel | CogSDE **(ours)** | 4,120,695 | 4,120,695 |
> > >
> > > Thus, the comparison isolates the effect of the transition model rather than encoder/decoder strength. Notably, even attention-based variants with slightly larger parameter counts (4,096,265) do not improve performance.
> > >
> > > We further clarify the limitation of latent AR in this setting: under long-horizon, command-driven prediction, errors accumulate at the transition level, especially under complex or stochastic command-driven dynamics. This likely contributes to the observed gap despite using the same decoder.
> > >
> > > Overall, this confirms that the performance gain of CogSDE is not attributable to model capacity, but to the latent dynamics formulation.
> > >
> > > ---
> > > **Q2: On the role of cross-attention.**
> > >
> > > We agree that cross-attention **does** interact with dynamics by modulating the autoregressive hidden state. Our point is that this interaction is **implicit and feature-level**, i.e., control influences hidden representations rather than directly shaping the transition law. However, in CogSDE the control enters **the latent transition law** explicitly.
> > >
> > > Specifically, CogSDE directly modulates the drift term with separate persistent and transient control channels, and injects uncertainty through state-dependent diffusion during rollout. This introduces a structured formulation where control and stochasticity are explicit components of the dynamics, rather than being absorbed into generic hidden-state updates.
> > >
> > > We believe this difference is important for the FM decoder: CogSDE produces a temporally coherent and structured latent trajectory, which serves as a more stable conditioning signal for generation. In contrast, autoregressive hidden states do not explicitly disentangle control effects and uncertainty, which may limit their effectiveness for long-horizon stochastic decoding.
> > >
> > > ---
> > > **Q3: Why explicitly separate observable motion and decision dynamics?**
> > >
> > > We **do not** claim that input-space SDE is infeasible; for moderate-dimensional motion, end-to-end modeling is absolutely possible. Our choice of a latent SDE is motivated by modeling structure.
> > >
> > > First, future motion in human and animal behavior is only partially observable: it depends not only on pose evolution, but also on **latent decision or intent states** that are not directly observed. In input-space models, these factors are entangled and must be captured **implicitly**, whereas the latent formulation makes this structure **explicit** and enables more stable long-horizon prediction under stochastic transitions.
> > >
> > > Second, this separation enables meaningful stability constraints: our boundedness analysis and dissipativity regularization act directly on the latent transition, which is harder to interpret in observation space due to measurement variability.
> > >
> > > Third, the latent formulation provides a cleaner interface for decoding, supplying a compact and structured trajectory that summarizes control and uncertainty, rather than requiring the decoder to operate on raw observations as dynamical states.
> > >
> > > We will revise the paper to clarify this as a structural modeling advantage rather than a strict necessity.
> > >
> > > ---
> > > We thank the reviewer for these questions, which help clarify key aspects of our work.

---

### Official Review · Reviewer_wfsF · 2026-03-13

**Soundness:** 3
**Presentation:** 2
**Significance:** 3
**Originality:** 2
**Overall Recommendation:** 4
**Confidence:** 3

**Summary:**

The paper proposes a control/action-conditional future prediction framework that uses a flow-matching generator conditioned on the latent representations of a discrete CDE model. The main novelty lies in conditioning the generator on the discrete CDE and in the resulting parameterization of the CDE components, including action-conditional terms, diffusion terms, and related dynamics. The model is built on top of the MoFLOW baseline (an unconditional flow-matching prediction codebase), extending it by integrating action-conditional and uncertainty-aware mechanisms through a latent CDE formulation. The proposed approach is evaluated against several baselines, including MoFLOW and transformer- and diffusion-based predictors. Additionally, ablation studies are conducted to disentangle the contribution of different components and determine which elements drive the overall performance improvements.

**Compliance With Llm Reviewing Policy:**

Affirmed.

**Final Justification:**

I thank the reviewers for the detailed responses. While my concerns have been addressed, I remain uncertain about this paper meriting acceptance. The method and tayloring SDEs for action-conditional predictions are potentially valuable for future development of more principled and interpretable approaches (world modelling, dynamics learning etc that go beyond the current application they have), even if the work is not yet state-of-the-art and contributions are incremental. I appreciate the authors' transparency about the limitations and trade-offs of their approach, and reward this accordingly by raising my score to a weak accept (with slight decrease in confidence score). That said, I will not champion the paper if other reviewers feel otherwise.

**Key Questions For Authors:**

1. The paper would benefit from a broader comparison against action-conditional latent dynamics models that operate on similar principles. Models like AC-RKN (Shaj et al., CoRL 2020), which achieves action conditioning through structured additive interactions in a Kalman-filter-inspired latent space with process noise acting as the diffusion term, or the Deep Kalman Filter (Krishnan et al., 2015) and its successors operate under discrete-time CDE-like assumptions in latent space and represent a natural comparison class. Similarly, the Dreamer series (Hafner et al., 2020, 2023) provides a strong world-modelling baseline with action-conditional latent dynamics. On the generative side, comparisons with diffusion/flow-matching approaches for action-conditional prediction/world mdoelling (you can severl based on simple search), would help situate the contribution. What specifically does the CDE formulation give you over these alternatives, and is this advantage demonstrated empirically? Is it the data-efficiency aspect or is some particular nature of the application domain?

2. Why a flow-matching generator, rahter than a simple MLP decoder? In many cases, the multi-modality in future trajectories is minimal for action conditional generators compared to open loop generators.

2. A key potential advantage of the proposed parameterisation, particularly the state-conditioned diffusion and noise terms, is the ability to produce calibrated uncertainty estimates: the model should be more uncertain precisely when its predictions are likely to be wrong. Did the authors run any experiments specifically evaluating predictive calibration — e.g., checking whether predicted uncertainty correlates with actual prediction error, or whether the model assigns higher variance in regions of high contact uncertainty or out-of-distribution dynamics? This would be a meaningful differentiator from simpler action-conditional baselines and would strengthen the motivation for the CDE parameterisation.

**Limitations:**

No detailed discussion on the limitations, and the authors can be upfront about these, if any.

**Strengths And Weaknesses:**

Strengths

1. Making flow matching / diffusion-based generative models action-conditional is a well-motivated problem with broad applicability — forward dynamics modelling, world models, model-based planning, and beyond.
2. While the contribution is a relatively straightforward extension, it is an elegant one: parameterising a CDE model to condition the generator is a clean and a principled design choice.
3. The disentanglement of action-conditional and diffusion terms, along with the accompanying ablations, is informative and helps isolate the contribution of each component.

Weaknesses

1. The contribution feels incremental. The neural CDE could plausibly be swapped out for any number of action-conditional future predictors, probabilistic or deterministic, and coupled with a flow-matching generator with similar effect. It is not clear why the CDE is the uniquely right choice here.

2. The implementation is difficult to follow from the main text alone. Key details, how the two models are combined, the loss formulation, and the discretisation scheme, are deferred to the appendix, making it hard to assess the technical soundness without hunting for them.

---

> ### Author Rebuttal · Authors · 2026-03-31
>
> We thank reviewer for the constructive feedback. Below we provide point-by-point responses addressing each concern, and we hope our clarifications resolve the reviewer’s questions.
>
> ---
> **W1 & Q1: Novelty & necessity.** The advantage of our approach is not data efficiency, but modeling partially observed, non-communicable decision processes in animal behavior, where latent cognitive states must be inferred. This leads to more reliable long-horizon predictions and better uncertainty modeling.
>
> We include additional comparisons (Table 1) with latent autoregressive and RSSM/Dreamer-style models. Our method consistently shows better average ADE over long-horizon rollouts of 60 frames, indicating that the improvements are not due to arbitrary architectural choices.
>
> **Table 1:** Comparison of behavior prediction performance across different encoders.
> |ID|Dataset|Encoder|ADE_min↓|FDE_min↓|ADE_avg↓|FDE_avg↓|
> |-|-|-|-|-|-|-|
> |1|rat|RSSM|6.21|8.42|16.47|30.04|
> |2|rat|Latent AR(GRU)|9.42|13.53|17.63|33.97|
> |3|rat|Latent AR(transformer)|9.28|11.59|21.37|37.35|
> |4|rat|CogSDE(ours)|**3.79**|**6.54**|**10.42**|**24.27**|
> |5|babel|RSSM|0.022|0.035|0.044|0.089|
> |6|babel|Latent AR(GRU)|0.060|0.105|0.123|0.323|
> |7|babel|Latent AR(transformer)|0.090|0.162|0.090|0.162|
> |8|babel|CogSDE(ours)|**0.018**|**0.032**|**0.042**|**0.086**|
>
> ---
> **W2: Implementation details.** We thank the reviewer and revise the presentation to make the pipeline self-contained. We add Fig.1 (https://anonymous.4open.science/r/adfhoaqhif/fig1.jpg) to illustrate the full computation, including input encoding, latent rollout, and flow-based generation, clarifying how the components are combined.
>
> Given history $x_{1:T_h}, u_{1:T_h}$, we encode context $C=\mathrm{Enc_{\mathrm{ctx}}}(x_{1:T_h})$ and initialize
> $Z_{T_h}=\phi(x_{1:T_h},u_{1:T_h})$. The latent trajectory is rolled out via Euler--Maruyama:
>
> $Z_{k+1}=Z_k+f(Z_k,u_k)\Delta t+G(Z_k)\sqrt{\Delta t}\,\epsilon_k.$
>
> The latent states are fused as
> $c_t=\mathrm{Fuse}(y_t,t,C,Z_t)$ to condition the flow decoder
> $v_\theta(y_t,t\mid c_t)$, and the trajectory is generated by:
>
> $y_{t+\Delta t}=y_t+v_\theta(y_t,t\mid c_t)\Delta t.$
>
> Training is performed using a trajectory-level regression objective:
>
> $\mathcal{L_{reg}}=\| x_{\mathrm{pred}} - x_{\mathrm{fut}} \|_F^2=\frac{1}{B} \sum\_{b=1}^{B}\frac{1}{A}\sum\_{a=1}^A\min_k\sum\_{t=1}^T
> \|x\_{\mathrm{pred}}^{b,k,a,t}-x\_{\mathrm{fut}}^{b,a,t}\|_2,
>  $
>
> which serves as a practical surrogate for the underlying generative objective. We additionally incorporate a dissipativity-inspired regularization term $\mathcal{L}_{\mathrm{bnd}}$ to stabilize long-horizon latent dynamics (see response to Reviewer~RZtT, W1,W6,Q1). These steps will be moved to the main text for clarity.
>
> ---
> **Q2: Why use a flow-matching generator.** We thank the reviewer for this question. We conducted additional experiments by replacing the flow-matching generator with a simple MLP decoder across different latent backbones (controlled SDE, latent AR, and RSSM) in Table 2, which highlights two key observations.
>
> First, while MLP decoders perform well on the BABEL dataset, which exhibits smoother and less stochastic motion patterns, their performance degrades significantly on the rat dataset. This reflects the inherently higher uncertainty and irregularity in the rat task, making it more challenging to model with deterministic or weakly expressive decoders.
>
> Second, MLP-based models tend to produce less diverse predictions, as reflected by lower avg-ADE/FDE. While this can sometimes improve deterministic accuracy, it limits the ability to capture multimodal future trajectories. In contrast, more expressive generative decoders are better suited for modeling such stochastic behaviors.
>
> **Table 2:** Comparison of prediction performance with MLP decoder on rat and BABEL
> |id|dataset|Encoder|Decoder|ADE_min|FDE_min|ADE_avg|FDE_avg|
> |-|-|-|-|-|-|-|-|
> |1|babel|rssm|mlp|0.019|**0.029**|**0.036**|**0.069**|
> |2|babel|latent_ar(GRU)|mlp|0.039|0.042|0.045|0.082|
> |3|babel|latent_ar(transformer)|mlp|0.046|0.060|0.051|0.091|
> |4|babel|CogSDE(ours)|moflow|**0.018**|0.032|0.042|0.086|
> |5|rat|rssm|mlp|5.85|7.90|16.01|29.65|
> |6|rat|latent_ar(GRU)|mlp|10.88|16.15|14.24|22.10|
> |7|rat|latent_ar(transformer)|mlp|8.02|10.13|**9.04**|**17.18**|
> |8|rat|CogSDE(ours)|moflow|**3.79**|**6.54**|10.42|24.27|
>
> ---
> **Q3: Predictive calibration evaluation.** We thank the reviewer for highlighting this important aspect. We evaluate uncertainty calibration by measuring the correlation between predicted uncertainty and prediction error.
> As shown in Table 3, uncertainty exhibits strong positive correlation with both ADE and FDE (Pearson r up to 0.83 at the timestep level).
>
> **Table 3:** Correlation between predicted uncertainty and prediction error.
> |Granularity|Target|Pearsonr↑|Spearmanρ↑|p-value|
> |-|-|-|-|-|
> |trajectory|ADE|0.646|0.553|<1e-50|
> |trajectory|FDE|0.682|0.560|<1e-50|
> |timestep|ADE|0.828|0.928|~0|
>
> ---

---

> > ### Author Rebuttal · Reviewer_wfsF · 2026-04-01
> >
> > Thank you for all the clarifications... I hve a few questions
> >
> > 1. How was AR- transformer and GRU made action conditional? Its well know that AR variants usually exhibit worse performance on multi step predictions due to error accumulation... most time series literature use a transformer trained to make direct multistep ahead prediction for a fixed long horizon... and perform way better .. Why was GRU autoregressive? An encoder decoder can do latent state prediction if trained with imputation loss / missing values during training. Could you elaborate on how these were integrated ?
> >
> > 2. Could the authors elaborate on uncertainty quantification experiments?? How is predictive uncertainty measured ??

---

> > > ### Author Response · Authors · 2026-04-02
> > >
> > > Thank you for the follow-up. We clarify how the GRU / Transformer baselines are made action-conditional, why we adopt open-loop autoregressive rollout, and how predictive uncertainty is quantified.
> > >
> > > ---
> > > **Q1: GRU / Transformer AR details**
> > >
> > > **(1) Historical motion and action conditioning**
> > >
> > > Given past motion $x_{1:T_h}$ and actions $u_{1:T_h}$, we encode motion using the same encoder as MoFlow:
> > >
> > > $h_x = E_{\mathrm{hist}}(x_{1:T_h}),$
> > >
> > > where $E_{\mathrm{hist}}$ is instantiated as a PointNet-style polyline encoder per agent followed by a Transformer encoder.
> > >
> > > The historical action sequence is summarized via a bidirectional GRU:
> > >
> > > $h_u = E_{\mathrm{ctrl}}(u_{1:T_h}),$
> > >
> > > and fused into a history context:
> > >
> > > $c = W_c[h_x;h_u],$
> > >
> > > which initializes the latent rollout.
> > >
> > > **(2) Future action conditioning in latent rollout**
> > >
> > > Future controls $u_{1:T}$ are injected step-by-step, where $z_0$ is initialized from the history context $c$ via a linear projection $z_0=W_zc$.
> > >
> > > #### **(2.1) GRU latent-AR baseline**
> > > At each step:
> > >
> > > (a). We first construct the input to the transition model by concatenation:
> > >
> > > $\xi_t = [z_{t-1};u_t;c].$
> > >
> > > (b). Then, the GRU hidden state summarizing past latent trajectory is calculated as:
> > >
> > > $h_t = \mathrm{GRUCell}(\xi_t, h_{t-1}),$
> > >
> > > where $h_0 = W_h c$.
> > >
> > > (c). The mean and log-variance of latent distribution are mapped by $g(\cdot)$, from $h_t$ and $c$ to Gaussian parameters:
> > >
> > > $(\mu_t,\log\sigma_t^2)=g([h_t;c]), \quad
> > > z_t \sim \mathcal{N}(\mu_t,\sigma_t^2 I).$
> > >
> > > Sampling is performed via the reparameterization trick:
> > >
> > > $ z_t = \mu_t + \sigma_t \odot \epsilon_t, \quad \epsilon_t \sim \mathcal{N}(0, I). $
> > >
> > > (d). **This implementation yields the autoregressive factorization**
> > >
> > > $p(z_{1:T}\mid u_{1:T},c)=\prod_{t=1}^{T} p(z_t\mid z_{<t},u_{\le t},c).
> > > $
> > >
> > > #### **(2.2) Transformer latent-AR baseline**
> > >
> > > We replace the recurrent transition with a causal Transformer operating on a growing token memory.
> > >
> > > At each step:
> > >
> > > (a). We first calculate the token embedding at step $t$, constructed from latent, action, and context:
> > >
> > > $e_t = W_{\mathrm{tok}}[z_{t-1};u_t;c].$
> > >
> > > (b). The initial token is $m_0 = W_{\mathrm{init}} c$, and the token memory up to step $t$ is:
> > >
> > > $M_t = [m_0, e_1, \dots, e_t].$
> > >
> > > (c). Then $H_t$ encoded token sequence after self-attention with positional encoding, and $h_t$ is the last-token representation summarizing history:
> > >
> > > $H_t = \mathrm{TransformerEncoder}(M_t;\text{causal mask}),
> > > \quad h_t = \mathrm{LN}(H_t^{\text{last}}).$
> > >
> > > (d). The latent distribution parameters are obtained in the same way as in the GRU baseline:
> > >
> > > $(\mu_t,\log\sigma_t^2)=g([h_t;c]), \quad z_t \sim \mathcal{N}(\mu_t,\sigma_t^2 I).$
> > >
> > > Sampling is performed via the same reparameterization trick:
> > >
> > > $ z_t = \mu_t + \sigma_t \odot \epsilon_t, \quad \epsilon_t \sim \mathcal{N}(0, I). $
> > >
> > > (e). Thus the Transformer baseline also satisfies:
> > >
> > > $p(z_{1:T}\mid u_{1:T},c)=\prod_{t=1}^{T} p(z_t\mid z_{<t},u_{\le t},c),$
> > >
> > > It is an action-conditional autoregressive latent model, differing from GRU only in how history is represented.
> > >
> > > **(3) Decoding to future motion**
> > >
> > > Given latent rollout $z_{1:T}$, future motion is generated via the same decoder interface:
> > >
> > > $\tilde{c}_t = \mathrm{Fuse}(y_t,t,c,z_t), \quad v_\theta(y_t,t \mid \tilde{c}_t).$
> > >
> > > All baselines share the same conditioning interface to the decoder and differ primarily in the latent transition mechanism.
> > >
> > > **(4) Why open-loop autoregressive rollout**
> > >
> > > We use open-loop rollout to match the information pattern of our method: predicting future latent states from past observations and future controls only. This ensures comparisons are aligned at the rollout level and isolates the effect of latent transition design.
> > >
> > > We agree that direct multi-horizon predictors are also strong baselines; they correspond to a different prediction protocol and are an important extension.
> > >
> > > ---
> > > **Q2: How is predictive uncertainty measured?**
> > >
> > > **(1) Predictive uncertainty**
> > >
> > > For each sample, we draw $K$ stochastic predictions $\hat{\mathbf y}_{i,k,t,j}$. The predictive mean is:
> > >
> > > $\bar{\mathbf y}_{i,t,j}=\frac{1}{K}\sum\_{k=1}^K \hat{\mathbf y}\_{i,k,t,j}.$
> > >
> > > We define uncertainty as the **Monte Carlo predictive variance across samples**:
> > >
> > > $U_{i,t}=\frac{1}{KJD}\sum\_{k=1}^K\sum\_{j=1}^J\left\|\hat{\mathbf y}\_{i,k,t,j}-\bar{\mathbf y}\_{i,t,j}\right\|_2^2,$
> > >
> > > $U_i=\frac{1}{T}\sum_{t=1}^T U_{i,t}.$
> > >
> > > **(2) Prediction error**
> > >
> > > Errors are computed from the mean prediction:
> > >
> > > $E\_{i,t}=\frac{1}{J}\sum\_{j=1}^J\left\|\bar{\mathbf y}\_{i,t,j}-\mathbf y\_{i,t,j}\right\|_2,$
> > >
> > > $\mathrm{ADE}\_i=\frac{1}{T}\sum\_{t=1}^T E\_{i,t}, \quad
> > > \mathrm{FDE}\_i=E_{i,T}.$
> > >
> > > **(3) Correlation analysis**
> > >
> > > We compute Pearson and Spearman correlations between:
> > >
> > > traj-ade: $ (U_i,\mathrm{ADE}\_i)$, traj-fde: $ (U_i,\mathrm{FDE}\_i)$, time-ade: $(U_{i,t},E_{i,t})$
> > >
> > > with timestep pairs flattened over all $(i,t)$. This uncertainty corresponds to empirical predictive variance induced by stochastic rollout.

---

### Decision · Program_Chairs · 2026-04-30

**Decision:**

Accept (regular)

**Comment:**

This paper tackles the question of long-horizon prediction using a controlled SDE framework.

- The problem addressed is important and the authors presents a technically solid story based on latent decision uncertainty in long-horizon prediction.
- The rebuttal was helpful to resolve many of the reviewers concerns (implementation details, quantitative comparisons, relation between theory and implementation) and lead them to increase their score.
- The proposed works has several strengths: broad applicability, elegant approach, compelling empirical and theoretical contributions)

Recommendation for the camera-ready version:
- Include a figure summarizing the paper.
- Move the full pipeline description into the main paper.
- Be explicit about the continuous-time vs Euler-Maruyama implementation gap
- Compact the uncertainty evaluation section